# Modelling Herbivory Impacts on Vegetation Structure and Productivity

Jens Krause[1], Peter Anthoni[1], Mike Harfoot[2], Moritz Kupisch[1], Almut Arneth[1,3]

1: Karlsruhe Institute of Technology, IMK-IFU, Campus-Alpin, Garmisch-Partenkirchen, Germany
2: Vizzuality UK, Gwydir St, Cambridge CB1 2LJ, United Kingdom
3: Karlsruhe Institute of Technology, IfGG, Karlsruhe, Germany

*Correspondence to*: Jens Krause (jens.krause@kit.edu)

**Abstract.** Animal herbivory can have large and diverse impacts on vegetation and hence on the state and function of ecosystems. Despite this, quantitative understanding of vegetation responses to consumption of green leaf tissue by herbivores is currently lacking. The large-scale impacts of changes in herbivore abundance on ecosystem function have yet to be investigated. Process-based modelling can help to quantify how animals affect important processes, such as ecosystem carbon cycling. To do so, we linked the dynamic global vegetation model LPJ-GUESS with Madingley, a model of multi-trophic functional diversity. This implementation allows us to simulate feedbacks between the availability of green vegetation biomass, herbivory and the whole trophic chain in response to monthly consumption of leaf biomass. In the coupled model system, we see an overall reduction in ecosystem productivity (NPP -5.2%), leaf area index (-9.0%) and carbon mass (-9.7%), compared to the stand-alone version of LPJ-GUESS, with the highest impact on carbon mass in the boreal ecosystems (-42%). We observe ecosystem composition to shift from boreal coniferous forests (without animals) to boreal mixed forests (with animals), as well as a general increase in herbaceous vegetation. Indirect effects like an increased light transfer facilitating growth of lower canopy layers are also captured by the model system. Overall, the results of this study underpin the important role of animals in ecosystem functioning and highlight the important contribution of process-based modelling towards a better understanding of complex food web interconnections.

## 1 Introduction

A number of review papers have highlighted animals' potential to notably affect local ecosystem functioning by altering canopy structure, productivity and biomass (Arneth et al., 2020; Cardinale et al., 2012; Schmitz et al., 2014, 2018; Sobral et al., 2017; Wilmers and Schmitz, 2016), but whether the highlighted interactions affect carbon-, nutrient-, and water cycles globally, and how these interactions may change in time, remains unclear. The scientific literature includes studies that have shown an increase in the abundance of herbivores can enhance ecosystem productivity by accelerating nutrient cycles (Enquist et al., 2020), reduce it through damaging plant individuals (Jia et al., 2018) or shift the distribution of plant species (Schmitz et al., 2014). Given the urgency of climate change, the importance of nature in combatting climate change and providing important contributions to people, and the ongoing changes in global biodiversity including size structured

defaunation, and efforts to reverse this loss through ecosystem restoration and rewilding (Schmitz et al., 2022, 2023), gaining a better understanding the role of animals for terrestrial ecosystem functioning is fundamentally important (Forest et al., 2023; Weiskopf et al., 2022).

Biogeochemical cycle models have largely omitted the influence of animals. Still, a limited number of local and regional-scale modelling experiments exist to investigate animal impacts on vegetation dynamics and nutrient cycling. For instance, Pachzelt et al. (2015) coupled a physiological grazer population model with a dynamic vegetation model. They found for African savannas net primary productivity together with precipitation to be the strongest predictor of modelled grazer densities but that vegetation biomass and burned area in turn were not affected substantially by varying grazer density. Similar results were found by Riggs et al. (2015), who coupled a landscape fire succession model with a multi-species herbivory module to study ungulate herbivory as driver for the emergent fire regime. Even though herbivory was found to play a role within single stands, across larger scales herbivory did not significantly impact respiration, primary production, carbon mass or the area burned. In contrast, Dangal et al. (2017) demonstrated with a combined mammalian herbivore population and land ecosystem model, that herbivores have a significant negative impact on net primary production and respiration. Berzaghi et al. (2019) incorporated statistical elephant disturbance in the Ecosystem Demography model and showed that introducing herbivores increases the long-term equilibrium above-ground biomass in the African rainforests but decreased the forest's net primary production.

So far, model studies on interactions between vegetation productivity and herbivores did not investigate how these reverberate to the entire trophic chain. Consequently, it is also unknown whether the model-based results – or observation-based evidence from experimental plots - can be generalised to larger regions, different environmental contexts, and how herbivore-omnivore-carnivore interactions would feedback on the biomass consumed. To our knowledge, the only globally applicable model of functional animal diversity is the Madingley model (Harfoot et al., 2014). Madingley has been shown to reproduce large scale ecological patterns alongside relationships in food webs (Hoeks et al., 2020), as well as being able to simulate small-scale field experiments (IOPscience, 2023). In Krause et al. (2022) we presented a version of the Madingley model that is driven by vegetation biomass simulated with a dynamic global vegetation model called Lund-Potsdam-Jena General Ecosystem Simulator (LPJ-GUESS) (Smith et al., 2014; Wårlind et al., 2014). The previous work explored how the amount and dynamics of green biomass available for consumption to herbivores affects the modelled trophic chain. This study aims to investigate the influence of multi-trophic food chains on ecosystem productivity, and to quantify whether the monthly herbivory affects canopy composition. This work is a next step towards a fully coupled modelling system that eventually will enable exploration of the effects of interactive vegetation, soil and animal processes on carbon and nitrogen cycling under present conditions, future climate change and under multiple other scenarios of anthropogenic influence.

**2 Methods**

We developed a coupled model system that consists of two independent models. Both modified models can run separately from another or are able to exchange data. For LPJ-GUESS, we based the model developments for this study on the LPJ-GUESS trunk version r10042 and implemented defoliation methods similar to Kautz et al. (2018). For Madingley, we based our development on the model code from Krause et al. (2022).

**2.1 The Madingley Model**

The Madingley model simulates plant-animal and animal-animal functional interactions, based on ecological principles and designed to predict global animal communities (Harfoot et al., 2014; Purves et al., 2013). We adapted the version by Krause et al,. (2022), coupling Madingley with a 0.5° grid resolution and a multi-year LPJ-GUESS climatology. To ensure spatial explicitness, we generated the LPJ-GUESS grid cell list during Madingley's initialisation phase.

Animals are modelled as cohorts; groups of individuals with shared properties. Categorical traits define feeding type (herbivore, carnivore, omnivore), reproductive strategy (iteroparous or semelparous), and thermoregulation strategy (endothermic or ectothermic), resulting in nine animal functional types (AFTs) - excluding endothermic semelparous animals, which are absent. Each group also has distinct assimilation efficiency and body mass limits. Cohorts also have continuous traits, which include juvenile and adult body mass, age, current body mass, abundance, and reproductive mass.

In Madingley, live vegetation biomass in each grid cell is represented by a wet matter evergreen and deciduous autotroph stock. Herbivores can consume up to 10% of said vegetation stock mass, which is a simplified assumption that takes into account their limited horizontal range, the vertical structure of plants, and selective feeding habits. At present the Madingley model does not specifically address vertical tree structure. The fact that all animals have access to the same amount of leaf biomass gives large animals an inherent advantage, as lighter animals typically have a higher vertical reach, while large animals are typically ground-dwellers.

All animals have a specific daily active time that depends on their thermoregulation strategy and the surrounding climate. During this time, herbivore will forage and carnivores search for prey within their grid cell. Carnivore predation likelihood is based on optimal predator-prey body mass ratios. Omnivores combine both strategies but with reduced carbon assimilation efficiency (65% instead of 80%), trading off between herbivory and carnivory. Non-predation mortality includes background (accidents, disease), starvation (when body mass falls too low), and senescence (age-related, applied only to adults). Starvation risk depends on current vs. maximum body mass. To combat starvation, all cohorts must meet their metabolic cost. This depends on their active time and is based on field and basal metabolic rates.

Adult individuals contribute surplus growth to a reproductive body mass pool. Reproduction occurs when said pool exceeds a certain threshold. Semelparous cohorts also sacrifice parts of their own body mass to bolster survivability of their offspring. Offspring cohorts inherit traits from their parent cohort, with stochastic variation in adult body mass.

The model includes migration in the form of two terrestrial dispersal types. Juvenile cohorts only disperse randomly (natal dispersal), with distance scaling with body size. Adult cohorts also disperse when reproduction is limited by low densities or when facing starvation-level resource scarcity (behaviour-mediated dispersal). After a dispersal event, the total number of

cohort in a grid cell can exceed a set limit. If so, similar cohorts are merged to manage computational load.

Further information on the model's mathematical representation of ecological processes and interactions between simulated animals can be found in Harfoot et al. (2014) and the corresponding supplementary material – both of which still being accurate descriptions.

## 2.2 LPJ-GUESS

LPJ-GUESS is a dynamic global vegetation model which combines the advantages of an individual-based growth model with a global process-based representation of carbon, water and nutrient cycling to simulate vegetation-soil-atmosphere dynamics (Smith et al., 2014; Wårlind et al., 2014). LPJ-GUESS is analogous to the Madingley model in that plant species that share key ecological traits are grouped into plant functional types (PFTs). The PFTs are defined by categorical traits such as bioclimatic preferences, photosynthesis pathways, lifeforms, leaf physiognomy, phenology and shade tolerance.

Cohorts within the woody PFTs, which share identical traits are distinguished by properties like carbon and nitrogen masses, age or height. A list of all simulated plant functional types with a selection of their characteristics can be found in Table S1.

Annual processes, like leaf, root and sapwood turnover, biomass allocation and growth, and mortality are simulated at the beginning of the year. Short-term processes like soil hydrology, stomata regulation, photosynthesis, plant respiration, decomposition and phenology are simulated on a daily basis.

Photosynthesis follows a modified Farquhar model (Farquhar et al., 1980; Haxeltine & Prentice, 1996). Plant growth is calculated as the surplus of the photosynthesis rate after subtracting maintenance and growth respiration costs, yielding a daily increment of net primary production (NPP), aggregating to an annual NPP. Based on that, plants regrow leaf carbon mass at the beginning of each year. Seedlings in LPJ-GUESS are handled separately from the patch PFTs and are not affected by our coupling.

LPJ-GUESS incorporates gap-model features (Bugmann, 2001) with stochastic disturbances and a 100-year return interval. Each grid cell is simulated with 30 patches to capture variability. We simulate natural vegetation without utilising LPJ-GUESS's capabilities to simulate crop growth, forest management or land-use change. To represent fire in the model, the integrated SIMFIRE module was used (Knorr et al., 2016).

## 2.3 The Coupled Feedback Link

Throughout this study, we refer to three distinct stages of the coupled model system: the "default," "offline," and "online" coupled versions. The "default" version represents the model code without any modifications and the "online" version is presented in this study. The "offline" version refers exclusively to the version of Madingley presented in Krause et al. (2022).

Typically, LPJ-GUESS processes the entire time series for a single grid cell before proceeding to the next. This approach is computationally efficient, as it minimises memory usage by focusing on one grid cell at a time. However, this structure poses a challenge for extracting information across the entire model domain on a monthly basis. While this limitation was not an issue when Madingley was forced with prescribed vegetation biomass (Krause et al., 2022), addressing it became essential for the "online" coupled version of the model system. To overcome this limitation, we switched the spatial and temporal simulation loops for the "online" coupled version based on the LPJ-GUESS version r10042. This modification allowed the system to handle the entire LPJ-GUESS model domain for a given timestep before progressing to the next timestep. This adjustment was critical for enabling interaction and synchronisation between LPJ-GUESS and Madingley for a coupled configuration (Figure 1).

The feedback loop on LPJ-GUESS's side is implemented similarly to an external herbivory module which is executed at the end of each month. Removal of leaf biomass throughout the year is not only having short-term effects such as increasing light transfer, but also long-term effects such as reducing potential future productivity through removing photosynthetic tissue. Carbon mass conservation in a grid cell remains unaffected by Madingley's dispersal process, as dispersal occurs after all other ecological processes have been applied and data exchange with LPJ-GUESS is completed. As such, dispersal is effectively implemented between Madingley time steps. To exchange information between the models, we expanded the file-exchange approach described in Krause et al. (2022). Detailed information about the formulations of the coupling interface and data aggregation can be found in the supplementary material.

**2.4 Model Spin-Up**

Both models need to run through an independent and a coupled spin-up phase. During the independent spin-up phase, Madingley uses its build-in vegetation model. Herbivores, omnivores and carnivores are initialised with the exact same biomass density. During the first years of spin-up, the interactions between the groups and the simulated physiological processes in Madingley rapidly shifts the cohort biomass density towards more realistic magnitudes.

Under initial conditions, LPJ-GUESS's vegetation succession commences with the establishment of different PFTs governed by their bioclimatic limits and light (nutrient and water) conditions at the forest floor (Smith et al., 2014). Succession typically starts with herbaceous PFTs, followed by fast-growing shade-intolerant tree species that are later succeeded by slow-growing shade-tolerant trees. 500 years have proven to be a sufficient timespan for LPJ-GUESS, which also captures spin-up of soil processes (Smith et al., 2014).

After both models complete the independent spin-up phase, the data exchange between the models is enabled and a coupled online spin-up phase of 500 years is initiated to equilibrate LPJ-GUESS vegetation with Madingley herbivory, and Madingley's trophic pyramid with LPJ-GUESS vegetation biomass. During the first 50 years of this joint spin-up phase, the percentage of leaf biomass reduction in LPJ-GUESS in response to Madingley herbivory is gradually increased so as to not kill off plant age-cohorts with initially unrealistically high herbivory rates. After 50 years, 100% of the herbivory reduction

is passed to LPJ-GUESS and the models are considered fully coupled at this timestep. An illustration of the coupled model system, including the different spin-up phases, can be found in Figures **1** and **2**.

## 2.5 Simulation Setup

We simulate impacts of animal herbivory for the European-African continents, as a model domain that captures a large variety of biomes and climates (20°E – 50°W, 35°S – 75°N). We used historical climate data input from the CRUJRA v2.1 dataset (Harris, 2020), and historic $CO_2$ concentrations from 1901 onwards (Meinshausen et al., 2017). During the spin-up phases, both models repeatedly cycle the climate data from 1901 to 1930, applying a standard $CO_2$ concentration of 296 ppm. LPJ-GUESS uses 30 patches per grid cell to ensure stochastic stability. Throughout the whole simulation, the nitrogen deposition was kept constant at 2 kg N ha$^{-1}$ yr$^{-1}$ (Smith et al., 2014).

The Madingley model was forced with a monthly average of the climate forcing used by LPJ-GUESS. Its model domain was seeded with 50 cohorts of each functional group, while the maximum number of cohorts allowed per grid cell was set to 500. This reduction compared to the maximum number of cohorts in Krause et al. (2022) accounts for the increase in computational demand given the larger model domain. All simulations include the same definitions of animal functional groups described in Krause et al. (2022)

## 2.6 Analysis

The simulation was run over the period 1901 to 2014. Changes in continental-scale net primary production (NPP), gross primary production (GPP), leaf area index (LAI) and vegetation carbon are presented as percentage increase or decrease of the "online" simulation when compared to the "default" LPJ-GUESS simulation. Second, we analyse the distributions of the dominant PFTs, as determined by maximal LAI, throughout the model domain. We chose LAI as metric for PFT dominance since the PFT with the highest LAI is also the PFT with the largest surface available for photosynthesis within a grid cell. To examine the impacts of vegetation-animal interactions in more detail, we further identified 10 locations, shown in Figure **4** and **5**. We arranged them from highest productivity (A1 & E1) to lowest productivity (A5 & E5). In addition, we compared heterotroph and autotroph biomass levels from the three different stages of the coupling implementation in Madingley (default, offline and online) against each other so quantify the impact of the coupling on simulated animal populations.

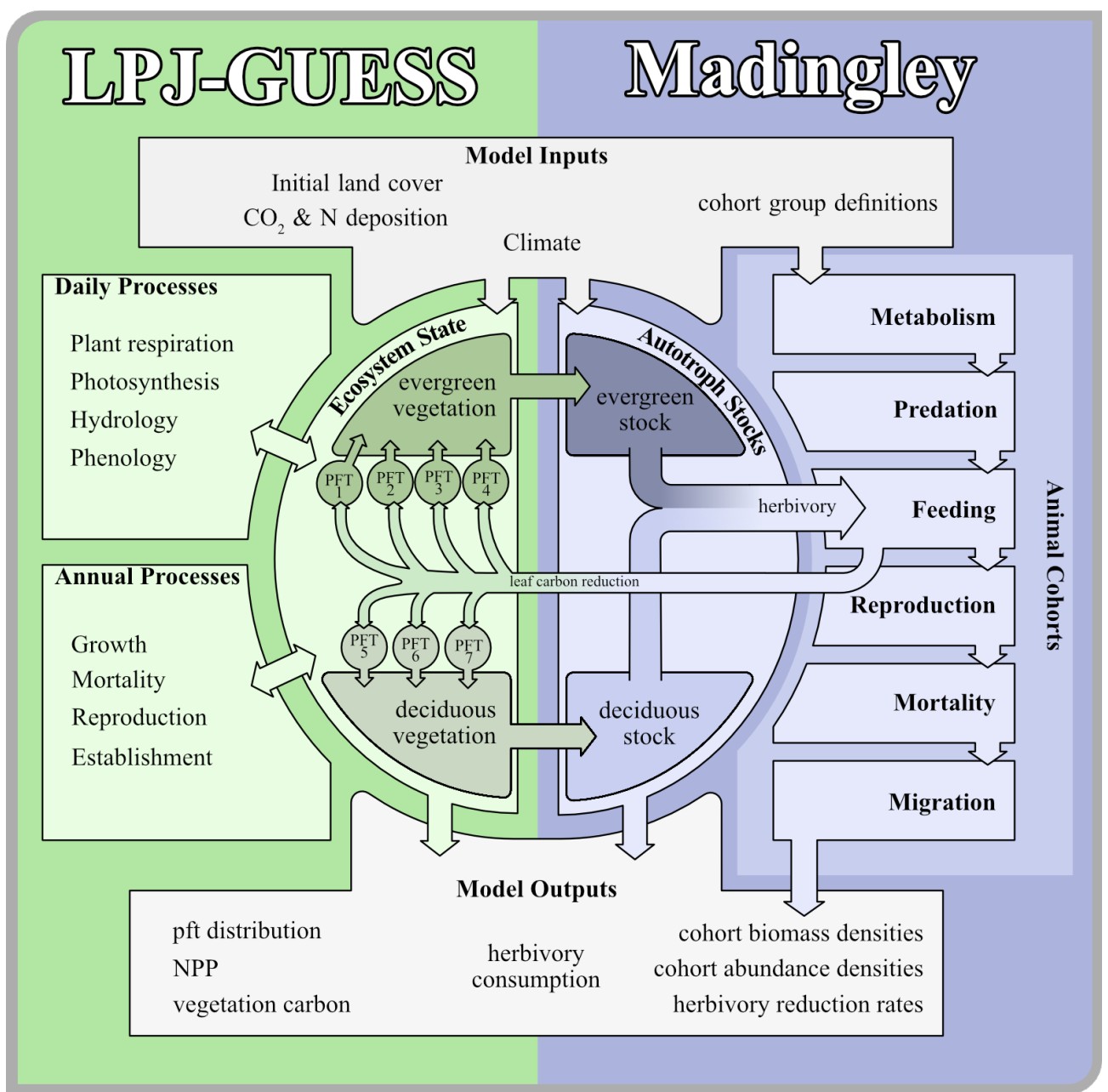

**Figure 1: Illustration of the coupling process. The panel shows how both models aggregate and exchange data and how the time loop is synchronised.**

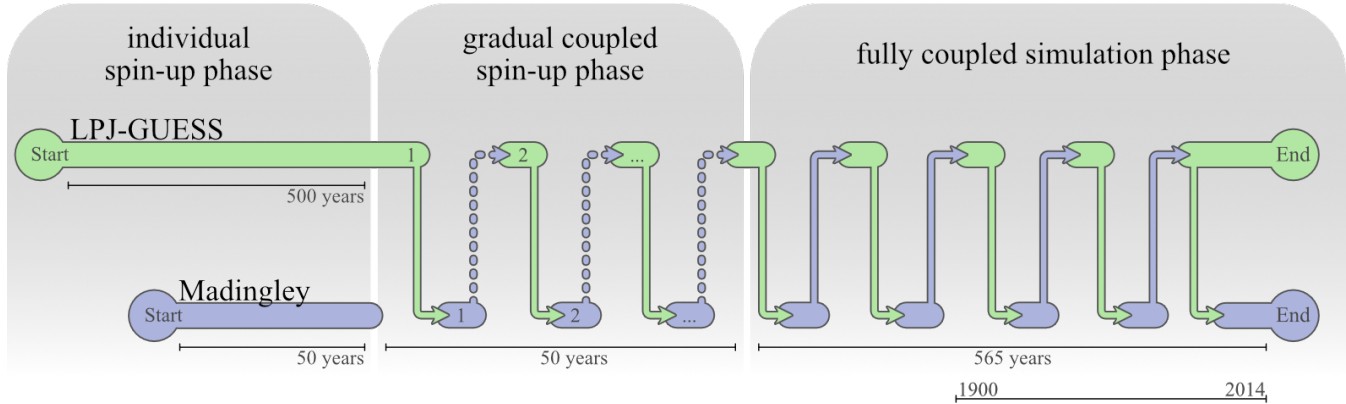

**Figure 2: Runtime structure of the coupled simulations. File transfer is indicated by arrows occurring after each month. First, LPJ-GUESS completes a timestep and passes data to Madingley, then enters a waiting state. Madingley subsequently runs the same timestep using the received data and returns herbivory data to LPJ-GUESS. The number of data exchanges shown is illustrative and does not represent actual model runtime or performance.**

We compared the model output of "default" and "online" LPJ-GUESS against monthly GPP flux estimates from the FLUXNET network and against satellite data, which are available via the International Land Modelling Benchmark Project (Collier et al., 2018) and are designed to be used for benchmarking ecosystem models. Since the FLUXNET measurement sites encompass the local site-climate but the simulation uses gridded 0.5° climate input, we preferentially aimed for regions within the simulation grid which include multiple FLUXNET stations and averaged the measured fluxes within the area. This was only possible in Europe, because there are only five FLUXNET stations in Africa. Thus, we compared FLUXNET station GPP data to the GPP of the corresponding simulated grid cell for three African stations. The FLUXNET dataset covers a time span from 1994 to 2014, but the individual stations only provide data for a fraction of the time span (see Figure **S4**). We averaged the available monthly GPP fluxes of the FLUXNET stations and compared them to the simulated average monthly GPP flux between 1994 and 2014.

We compared the simulation output to GPP data from the FLUXCOM dataset (Tramontana et al., 2016), LAI from the AVHRR dataset (Fang et al., 2019), evapotranspiration from the GLEAMv33a dataset (Miralles et al., 2011) and woody vegetation carbon from a combination of multiple datasets (Pugh et al., 2023; Saatchi et al., 2011; Thurner et al., 2014). The underlying data is of a much higher resolution than the 0.5° resolution of LPJ-GUESS, but the International Land Model Benchmarking project maintains a collection of datasets aggregated to 0.5°. Besides the AVHRR dataset, the datasets contain additional uncertainty estimates based on human impact factors like forest management and land use. The AVHRR data is also a model product and includes human land use. Forest vegetation (above- and below-ground) carbon estimates in Thurner et al. (2014) covers northern boreal and temperate forests, including the European region of our model domain for which Thurner et al. (2014) indicate low-medium uncertainty in their estimates. The data was compared to the simulated vegetation carbon from 1980 to 2000. For tropical forest and savanna total biomass estimates, we used the African fraction of the Saatchi et al. 2011 dataset (35°N - 35°S, 20°W - 50°E) which covers the time period 1995-2005. They estimate a relatively high (30-45%) uncertainty, especially in the tropical rainforest. The impact from human timber extraction is

210 included in the estimates derived from remote-sensing information, while we simulate potential natural vegetation, including a simplified estimates of natural disturbances and wildfires. Pugh et al. (2023) provided data that combined remotely sensed disturbance-intervals with LPJ-GUESS derived vegetation carbon for the years 2001-2014. To minimise human influence in our comparison, we chose the low disturbance natural vegetation scenario from Pugh et al. (2023), in which they focussed on protected areas to keep human influence in their estimates to a minimum. We also compared our simulations to a dataset of

215 aboveground biomass on non-cropland and non-pasture vegetation i.e. present land use distributions. It is based on the ESA 100m aboveground biomass dataset (Santoro and Cartus, 2023) and the ESA 300m land cover dataset (Version 2.0.7). For the comparison to our simulations, we assumed that aboveground biomass represents 70% of the total vegetation biomass, since LPJ-GUESS does not explicitly model aboveground biomass. Finally, we compared power-law relationships between NPP, herbivore biomass and consumption by primary consumers from each model version against similar relationships

derived by Cebrian (2004).

To evaluate the influence of key model assumptions, we conducted a sensitivity analysis focusing on two parameters expected to have the greatest impact on simulation outcomes: (1) the proportion of biomass edible to herbivores and omnivores, and (2) the fraction of evergreen leaves transferred to Madingley. Simulations were performed at three ecologically distinct locations: AF1 (high productivity), AF3 (non-seasonal with strong animal impact), and EU2 (seasonal

with strong animal impact). At each site, we ran an ensemble of three control simulations based on our baseline assumptions over a 5×5 grid cell domain. For each sensitivity test, one assumption was altered while the rest remained unchanged, and the ensembles were re-run. In total, we conducted 54 simulations and quantified the resulting effects using Hedge's d and percentage changes in average NPP, leaf carbon, and herbivore biomass. An experiment where not 100%, but only 10% of the eaten carbon and nitrogen is transferred to the LPJ-GUESS pools (assuming that the rest is respired by the animals) is

presented in the supplementary material.

# 3 Results

## 3.1 Productivity and Carbon Masses

Figure **3** shows the coupling-induced differences of simulated NPP on a grid cell level. Aggregated across the model domain, these changes are relatively minor when comparing results from LPJ-GUESS with herbivory (27.5 ± 1.4 Pg C) to those without herbivory (29.0 ± 1.4 Pg C). Herbivory, therefore, causes an overall reduction in NPP of 1.5 % Pg C (-5.2% ± 5.1%). However, substantial spatial variability is evident across the model domain. NPP generally increases in response to herbivory across most of the African continent but decreases in large parts of Europe.

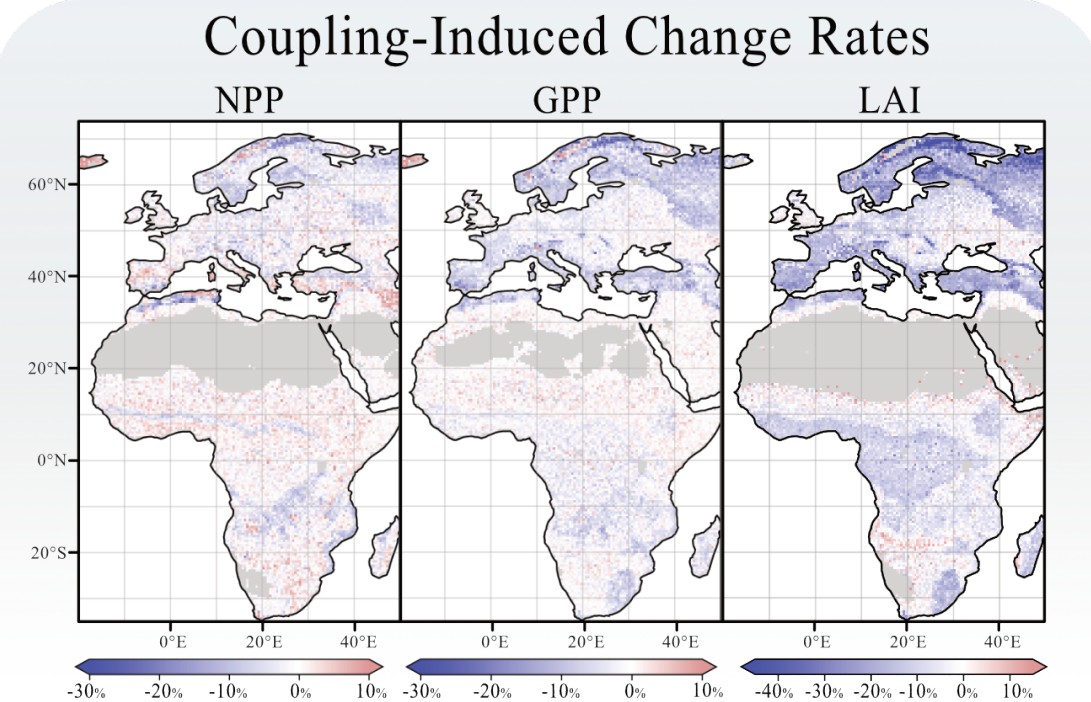

**Figure 3: Coupling related changes in grid cell's NPP, GPP and LAI. Each panel shows the percentage-wise difference caused by the coupling. All figures were rendered over the last 30 years of the simulation.**

The coupling responses in LPJ-GUESS are also reflected in changes to tree carbon mass. The effects of herbivory on total vegetation carbon are more pronounced than on NPP. Figure **4** shows this response across the model domain and provides a more detailed picture of the different PFTs at the ten selected sites. Across the model domain, vegetation carbon decreases by -9.7% (± 17.3%) in response to herbivory. The most significant reductions in vegetation carbon are found in the boreal regions, with, e.g., a decrease of -41.8% (site E4) and in savannas near grasslands, where vegetation carbon decreases by -37.5% (site A5). In northern Norway, however, there is a narrow expanse of boreal grassland that experiences a significant

increase in vegetation carbon. Besides this narrow expanse, the most prominent increases in vegetation carbon in response to herbivory are found in temperate mixed forests of central Europe (e.g., site E1, +3.9%).

Across the model domain, introducing herbivory leads to an average decrease in LAI of -9.0% (±10.1%), with the largest reductions occurring in boreal ecosystems. The spatial response of GPP closely mirrors the LAI response (Figure **3**), although the magnitude of change is less pronounced for GPP with an overall decline of -2.4% (±4.4%). Introducing herbivory to LPJ-GUESS results in a slight overall increase in evapotranspiration rates, averaging +0.2% (±4.2%). The most notable increases are observed in central Europe and tropical rainforest regions. At the individual sites, significant reductions in evapotranspiration were found of -6.5% at site E5 and -6.3% at site A5.

### 3.2 Canopy Composition

The changes in tree carbon mass arise not only from reductions in leaf carbon but also from alterations in PFT composition. Figure **5** shows each grid cell's dominant plant functional type, i.e. the PFT with the highest leaf area index, as well as the different PFTs contributions to the overall grid cell's LAI for the ten selected sites. In eastern Europe, woody vegetation shifts towards vegetation dominated by C3 herbaceous PFTs, represented by site E2. When exposed to herbivory, the contribution of C3 grass (C3G) to the total LAI increases markedly, from 6.1% without herbivory to 35.5%, making it the dominant PFT. This pattern is even more pronounced in the coldest ecosystems, represented by site E4. Here, the contribution of C3G to the total LAI rises sharply from 18.3% without herbivory to 50.9% with herbivory.

Patterns of PFT dominance can also be associated with positive or negative vegetation carbon responses. In the boreal regions of northern Europe, large decreases in vegetation carbon in response to herbivory are observed in grid cells dominated by boreal needleleaf evergreen (BNE) trees. In contrast, central Europe, predominantly characterised by temperate broadleaf summergreen (TeBS) trees, exhibits an overall increase in vegetation carbon in response to herbivory. A comparable positive response is also evident in landscapes dominated by grassy PFTs, particularly the boreal grassland in northern Norway. African grassland ecosystems, dominated by C3G or C4 grass (C4G), also show an overall increase in vegetation carbon.

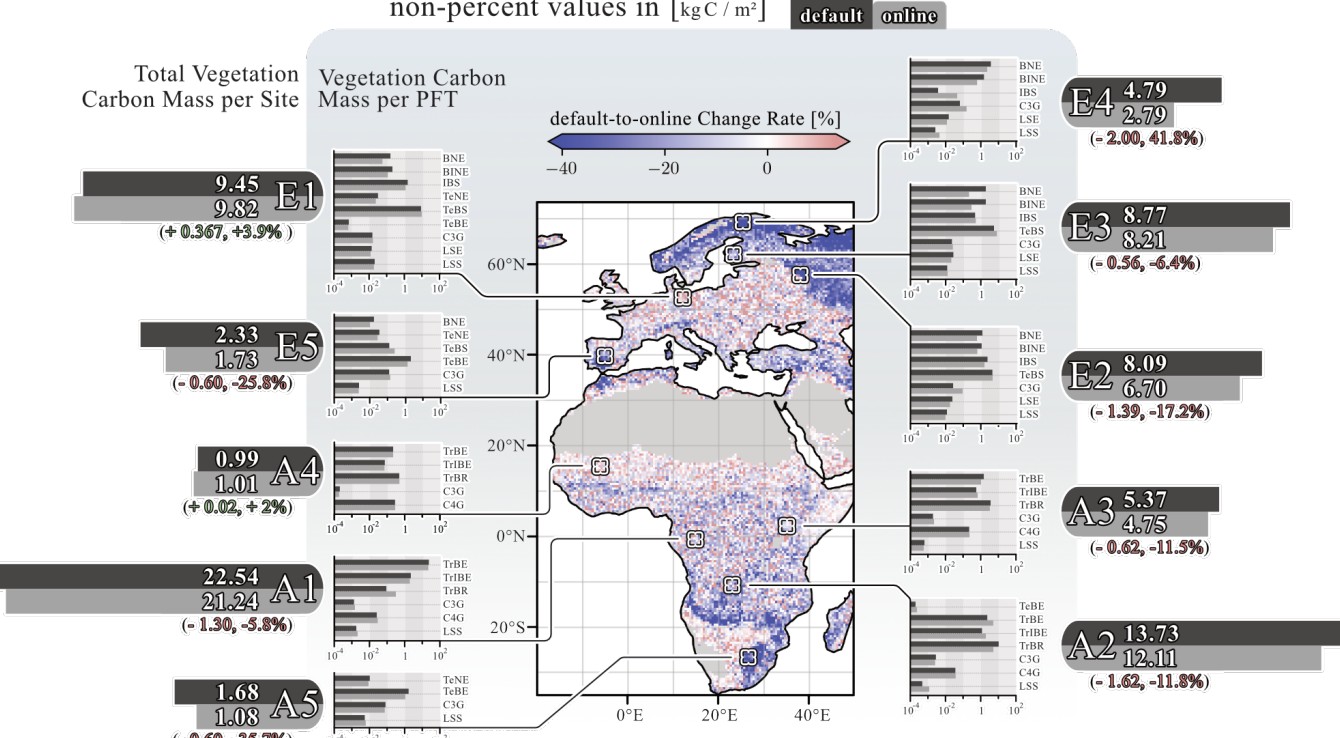

**Figure 4: Coupling-related changes in vegetation carbon for the simulation domain. The map illustrates the percentage differences between the "default" and "online" simulations per grid cell. Flanking the map on both sides are detailed vegetation carbon responses for the ten selected locations. The large horizontal bars next to the location tag represent the overall vegetation carbon response at that site, with its length corresponding to quantitative carbon levels. The smaller bars indicate the response of each PFT present at each location. Non-percent values of vegetation carbon are given in kg C m⁻². The figure reflects data averaged over the final 30 years of the simulation.**

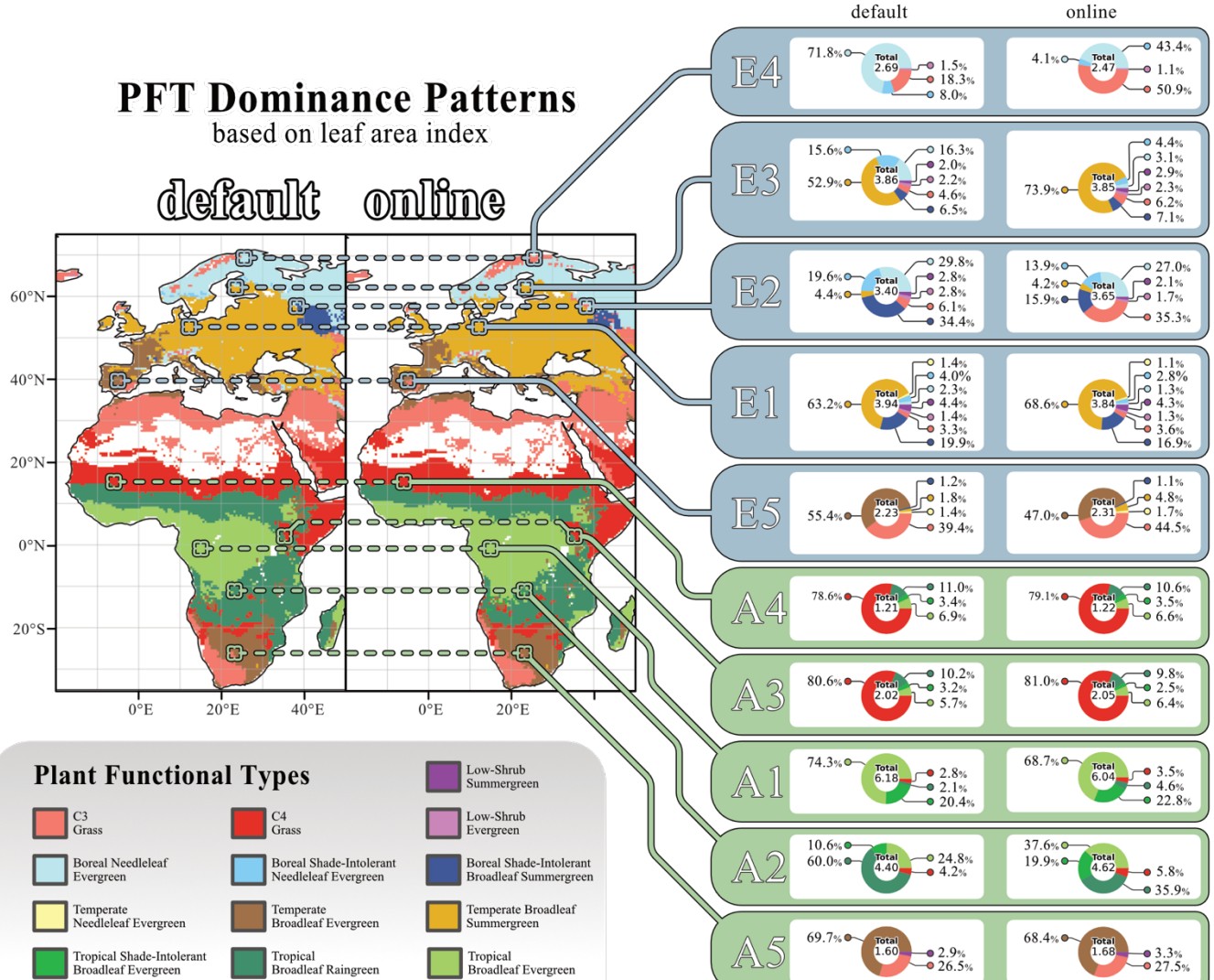

**Figure 5: Coupling-related changes in grid cell's dominant PFT. The maps display the dominant PFT in each grid cell, determined by the PFT's LAI in the "default" and "online" simulations. Only grid cells with a total LAI greater than 1 were included in the analysis. Flanking the maps, pie charts show the contributions of all PFTs to the total LAI for ten selected locations. The "default" simulation results are presented on the left, while those from the "online" simulation are shown on the right. The figure reflects data averaged over the final 30 years of the simulations. PFT Colour coding is labelled underneath the figure.**

### 3.3 Animal Populations

Differences in simulated leaf, herbivore, omnivore and carnivore biomass densities are compared across three stages of model development (Figure **6**): "default" Madingley, which uses the Miami model for vegetation; "offline" Madingley, described in Krause et al. (2022) and "online" Madingley. In "online" LPJ-GUESS/Madingley, significant reductions are evident for herbivore (-58% ± 47%) and carnivore biomass densities (-59% ± 36%). Omnivore biomass density decreases to a lesser extent (-19%). These substantial declines stem from LPJ-GUESS simulating less leaf biomass available for

consumption compared to "default" Madingley (-49% ± 32%). The high standard deviations are mostly caused by low-productivity regions adjacent to deserts, where "default" Madingley significantly overestimates annual NPP. Conversely, in areas where LPJ-GUESS simulates higher annual NPP than "default" Madingley, biomass densities of all animal groups increase.

Figure **6** also compares biomass densities for "default", "offline", and "online" Madingley based on the four sites investigated in Krause et al. (2022). For "offline" Madingley simulations, herbivore biomass increased at all four sites compared to "default" Madingley. In contrast, "online" Madingley shows lower herbivore biomass densities at all four sites compared to "default" Madingley. This reduction is especially pronounced at both European sites, where carnivore biomass densities are also significantly reduced in "online" Madingley. At the African sites, herbivore, omnivore and carnivore biomass densities in the "online" version are similar to those of "default" Madingley. However, "offline" Madingley simulates much higher biomass densities for all three groups.

Finally, Figure **6** shows biomass densities at three selected sites. The first site is located in the tropical rainforest, where LPJ-GUESS predicts the highest NPP. At this site, leaf and AFT biomass densities are similar between "default" and "online" Madingley. The second site is a grassland ecosystem adjacent to the Sahara Desert, where "online" Madingley simulates significantly lower leaf biomass densities (-71%) compared to 'default' Madingley. As a result, biomass densities for all AFTs are also significantly reduced, with the most notable decline in carnivores (-93%). The third site is located in Sweden, near the Baltic Sea, where "online" Madingley simulates higher leaf biomass densities (+17%) compared to "default" Madingley. This increase leads to higher omnivore biomass densities (+21%) and carnivore biomass densities (+34%), while herbivore biomass densities remain largely unchanged (-0.4%).

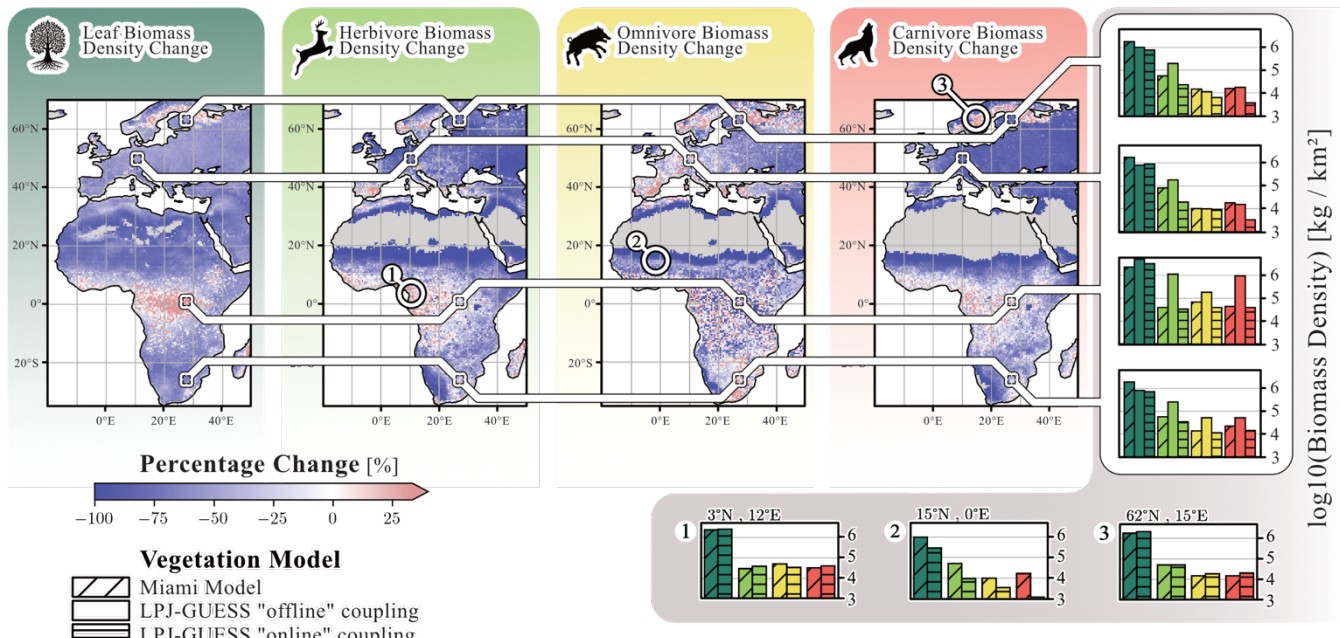

**Figure 6: Coupling-related changes in AFT biomass densities. The four maps show differences in leaf, herbivore, omnivore, and carnivore biomass between the "default" version of Madingley and the "online" coupled version. Additionally, the figure shows a comparison of the "default", the "default" coupled (Krause et al., 2022) and the "online" coupled Madingley versions at the four representative locations from Krause et al. (2022). Finally, we selected 3 locations that exhibit significant differences in biomass density levels among herbivores, omnivores and carnivores between the "default" and "online" versions: (1) represents a tropical rainforest biome, (2) represents a hot arid grassland biome and (3) represents a boreal grassland biome. The figure reflects data averaged over the final 30 years of the simulation.**

### 3.4 Evaluation

Outputs from the "default" and the "online" versions of LPJ-GUESS are compared to GPP derived from FLUXNET sites (Reichstein et al. 2007). Modelled monthly GPP fluxes are well within the standard deviation range of the measured GPP fluxes for both "default" and "online" LPJ-GUESS (Figure 7). However, in boreal areas 1, 2 (which correspond to boreal forests in Finland and Sweden) and area 5, simulated GPP tends to be higher than GPP derived from the flux towers in these regions.

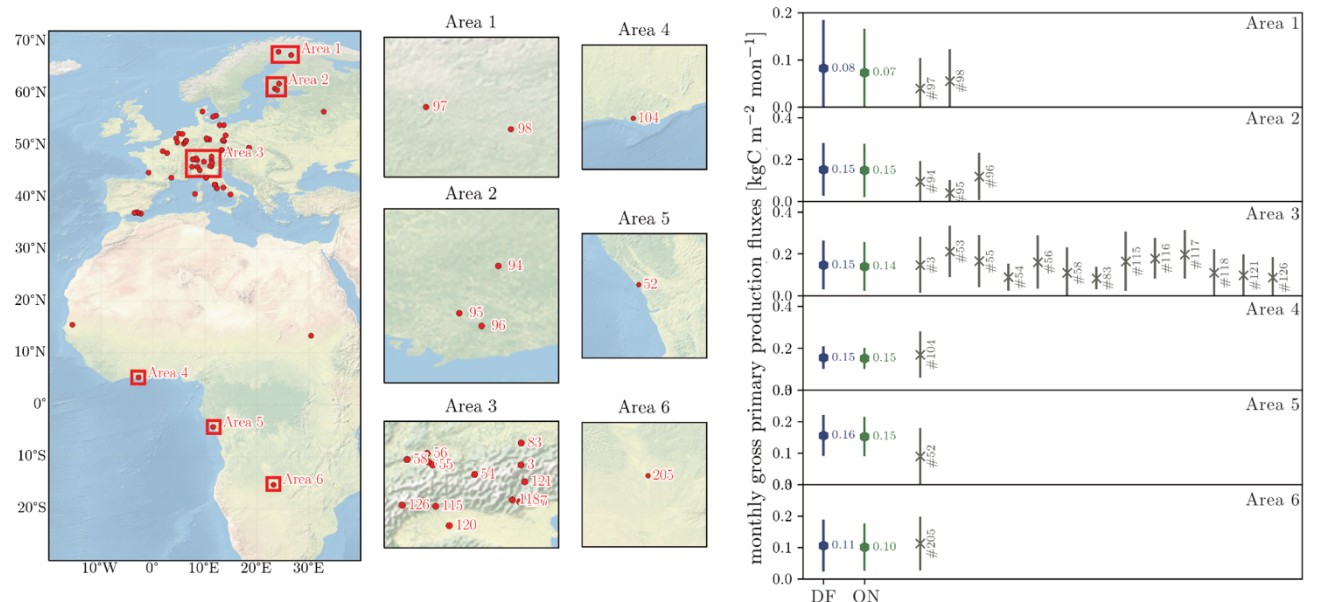

**Figure 7: (Left) Distribution of FLUXNET eddy covariance measurement sites over the model domain with selected Areas and station numbers. (Right) Comparison between "default" LPJ-GUESS (DF), the "online" coupled simulation (ON) and the FLUXNET measurements with station number. The displayed GPP fluxes are monthly fluxes from the selected area which contains the FLUXNET sites. Simulated GPP are for the period 1994 to 2014. The time spans of the corresponding eddy flux data (within that same period is provided by the separate FLUXNET stations and can be found in Figure S4. Background map "Shaded Relief and Water large 3.2.0" by Natural Earth I (download under https://www.naturalearthdata.com/downloads/10m-natural-earth-1/10m-natural-earth-1-with-shaded-relief-and-water/).**

When examining evapotranspiration and GPP across the model domain, regression lines for "default" and "online" LPJ-GUESS show little deviation (Figure **8**). The "online" version appears to be simulating GPP estimates that are somewhat closer to the measured values, particularly in boreal ecosystem grid cells. The evapotranspiration regression lines show no significant differences between "default" and "online" LPJ-GUESS. However, a slightly reduced slope is observed, reflecting the minor overall decrease in evapotranspiration. Both versions of LPJ-GUESS also demonstrate a tendency to overestimate LAI, especially in warm ecosystems such as those in southern Africa, where the satellite-derived LAI values range between 2 and 3, while simulated LAI is nearly doubled. However, the "online" version shows a lower discrepancy between simulated and satellite-derived GPP (Figure **8**).

A general issue with the comparisons presented above is that we compare data derived from measurements that include human-influenced land covers with simulations of natural vegetation. For many variables such as GPP or NPP (and to some degree also LAI), croplands and forests do not necessarily differ hugely such that the comparison is qualitatively useful, in particular since the purpose is to demonstrate that the coupling with herbivory does not push the model into an unrealistic state.

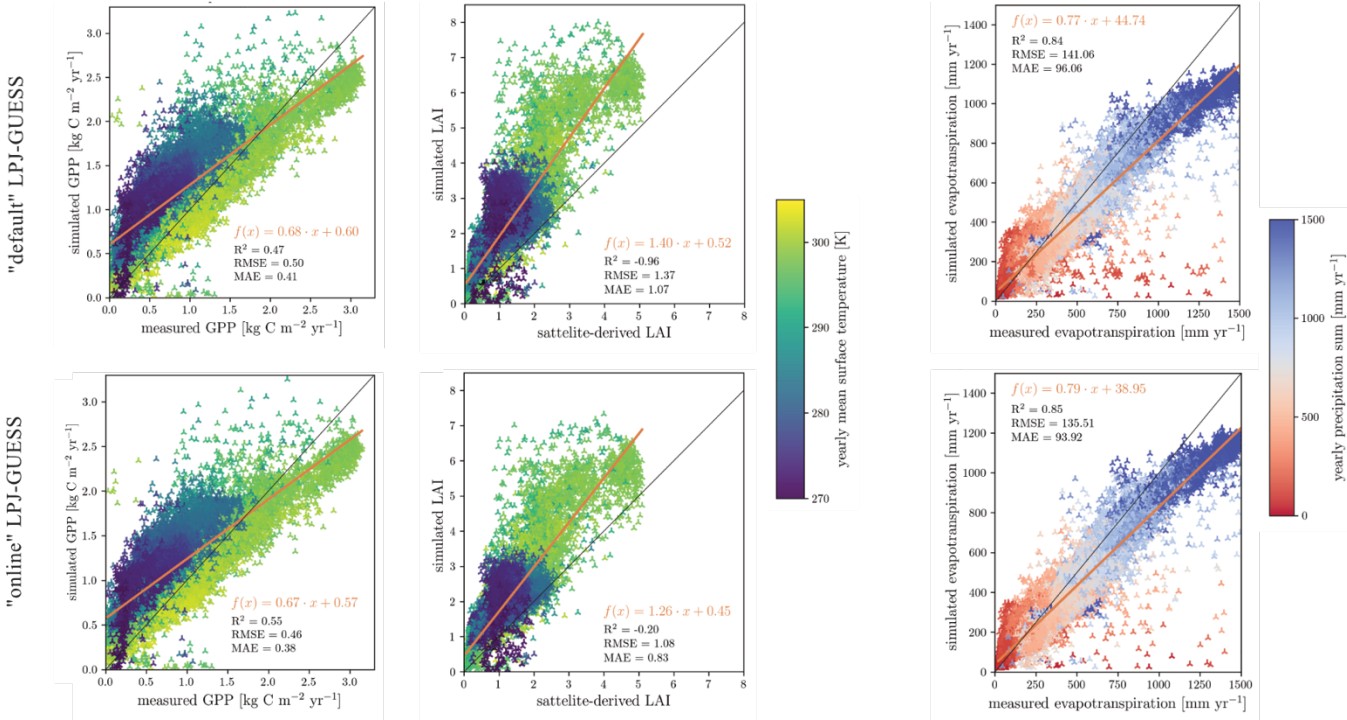

**Figure 8: Left)** Linear fit of simulated and measured GPP after FLUXCOM dataset (Tramontana et al., 2016). **Middle)** Linear fit of simulated and measured LAI after the AVHRR dataset (Fang et al., 2019). **Right)** Linear fit of simulated and measured evapotranspiration after the GLEAMv33a dataset (Miralles et al., 2011). All figures used model output between 1980-2014. Each marker shows the average of a grid cell over the 34-year timespan. The markers colour indicates the grid cell's yearly average near surface temperature and precipitation sum.

For carbon mass, the products we use here are for forests only and also include managed forests (Figure **9**). Overestimation of these data by our simulations is thus expected, as seen in particular in the comparison to Saatchi et al. (2011) for the forest vegetation carbon mass in Africa. For European forests, the dataset from Pugh et al. (2023), which attempts to be as close to an undisturbed state as possible, matches the modelled vegetation carbon best for both model versions. Compared to the non-cropland and non-pasture biomass (AGB) levels, based on the ESACCI from Santoro and Cartus (2021), the "default" and the "online" versions of LPJ-GUESS both show an overestimation in areas with low aboveground biomass and an underestimation in areas with high aboveground biomass.

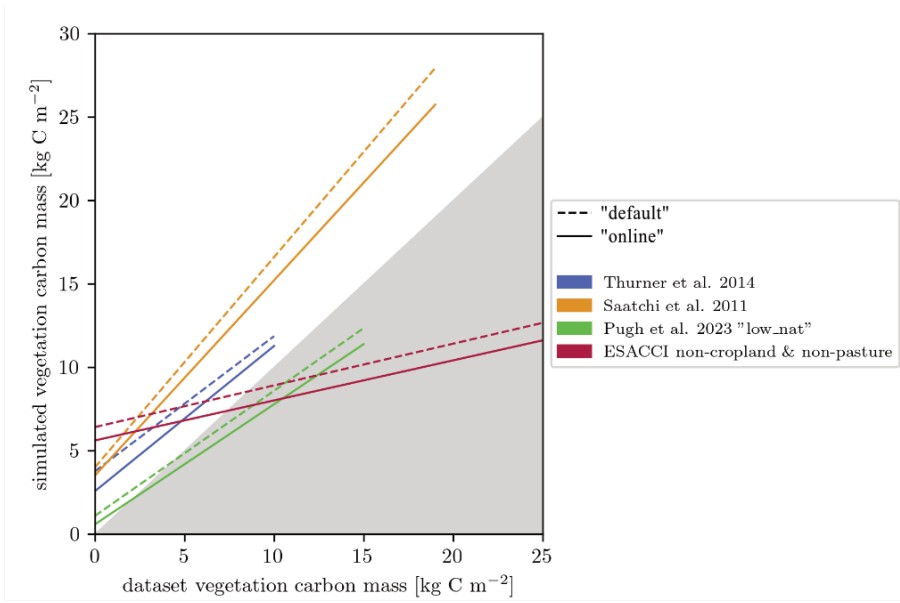

**Figure 9: Linear fit of simulated and measured vegetation carbon mass for four datasets. The non-cropland & non-pasture dataset is based on the ESA aboveground biomass (Santoro and Cartus, 2023) and the ESA land cover datasets (ESA Land Cover Climate Change Initiative (Land_Cover_cci): Global Land Cover Maps, Version 2.0.7, 2024). AGB was converted into total vegetation carbon mass.**

Figure **10** compares herbivore biomass levels and herbivory consumption between the "default" and "online" versions of the Madingley model, illustrating the relationship between grid cell herbivore biomass density (and herbivory consumption) and grid cell NPP. In the "default" Madingley model, herbivore biomass exhibits minimal variation with grid cell NPP, in contrast to the "online" version. This results in a power-law relationship with a steeper slope for the "online" model, aligning more closely with the power-law derived by Cebrian (2004) compared to both the "default" and "offline" versions (Figure **10**). However, the general response of herbivore biomass to underlying grid cell NPP is still overestimated by one order of magnitude for high-productive ecosystems and by two orders of magnitude for low-productive ecosystems.

We previously demonstrated that "online" coupling reverses the increase in animal biomass observed with "offline" coupling (Figure **10**). However, this reversal does not affect the emergent power laws, as the underlying vegetation also undergoes significant reductions.

While the herbivore biomass to NPP relationship varies significantly across model versions, the herbivory consumption rate to NPP relationships for all versions are similar to the relationship described by Cebrian (2004).

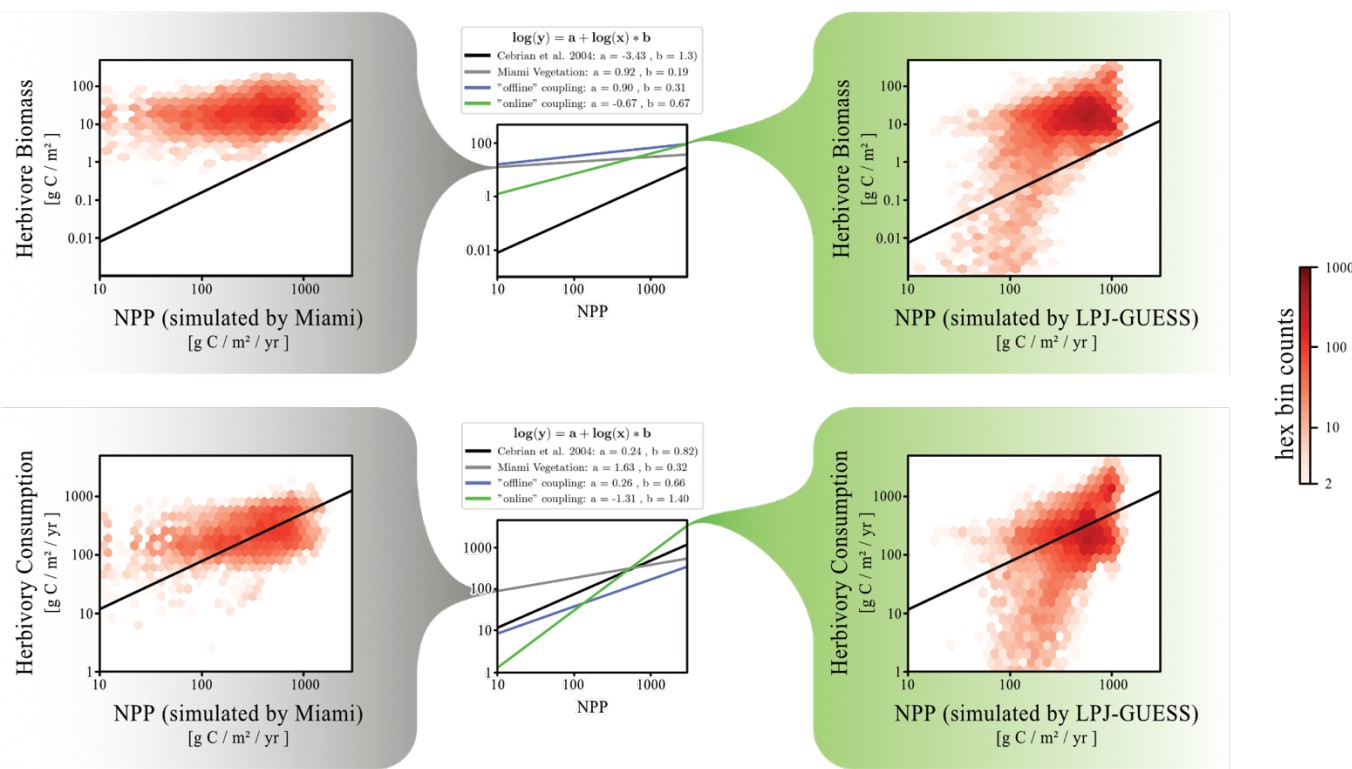

**Figure 10: Power-law relationship between herbivore biomass and grid cell productivity, as well as herbivory consumptions and grid cell productivity. The red saturation of each hexagon represents the number of data points found in the model system that matches the NPP and the herbivore biomass (or herbivory consumption) of that hexagon's position in the grid. The left distributions show data from "default" Madingley (driven by the Miami model), while the right distributions show data from "online" Madingley. In the middle of the figure, power-law relationships are summarised for all three Madingley model versions.**
**The black line in each plot displays the power-law derived by Cebrian (2004).**

## 4 Discussion

### 4.1 About the impact magnitude of animals on modelled vegetation

A high abundance of herbivores often results in reduced vegetation biomass (Dangal et al., 2017; Jia et al., 2018; Schmitz et al., 2014; Staver et al., 2021). In the current version of the coupled LPJ-GUESS/Madingley model system, no other effects
like the acceleration of the nutrient cycles (Enquist et al., 2020) or shifts in plant species distribution through selective feeding (Schmitz et al., 2014; Staver and Bond, 2014) are parameterised. Thus, the general trends we see in vegetation and process-responses are caused by the monthly removal of green biomass. This removal in principle should reduce canopy photosynthesis, although the effect may be partially compensated by more light reaching into lower layer of the canopy. We, therefore, expected a reduction of plant biomass as a response to the herbivore introduction. While the overall effects across
the modelled domains are small, spatial variation in the impacts are visible, which indicate biome specific dynamics. Nevertheless, the overall response is showing a negative trend in productivity, vegetation biomass and LAI as we expected.

The spatial patterns we found in our simulations show that under boreal climate conditions, the impact of leaf damage on photosynthesis may be more noticeable than in warmer. Boreal ecosystems are characterised by a shorter growth season with fewer days that allow high photosynthesis rates. When leaves are damaged by herbivores or omnivores, this limitation becomes even more constraining. The reduced photosynthesis rates cannot be compensated by an increased light transfer to deeper canopy layers. In the modelled boreal ecosystems, omnivores gain a competitive advantage over herbivores by flexibly adapting their diet in response to limited leaf biomass availability. This advantage is reflected in the increased omnivore biomass density observed in the "online" model version compared to the "default" Madingley setup (see Figure **6**), while herbivore and carnivore biomass densities decline. The rise in omnivore abundance further contributes to the reduction of leaf biomass, as omnivores require greater biomass intake than herbivores due to their lower assimilation efficiency.

Changes in ecosystem productivity were previously described by studies investigating general empirical relationships between herbivores and ecosystem productivity and structure: Dyer et al. (1993) reported that low herbivory pressure can enhance productivity while high herbivory pressure leads to reductions in ecosystem productivity. The authors also emphasized that the reported effects vary considerably across space and time. Schmitz et al. (2014) argued that in boreal ecosystems, leaf damage caused by moose can cause declines in $CO_2$ uptake and storage directly by browsing on photosynthetic tissue and indirectly through supressing tree growth. In other boreal ecosystems, moose were reported to lower primary productivity of tree species (Bonan, 1992; Kielland and Bryant, 1998; Schmitz et al., 2003). Another aspect of the herbivory response in central and northern Europe is that broadleaf summergreen PFTs are becoming more prominent. In reality, evergreen plants grow leaves that are less nutritious than leaves of deciduous plants and animals chose their nutrition source depending on the nutrient content (Villalba and Provenza, 2009). In contrast, animals modelled by the Madingley model attempt to meet their metabolic cost through consuming evergreen and deciduous plants without taking leaf C:N ratios into consideration. The impact observed through e.g., moose selectively browsing on deciduous vegetation is not yet included. Still, our modelled responses broadly go into the right direction although at the moment we expect animals to overprioritise evergreen vegetation in our coupled model system.

For the African continent, both LPJ-GUESS versions are overestimating productivity, LAI and vegetation carbon mass. Staver and Bond, (2014) found that large browsers are likely to suppress tree establishment in savannas by damaging young trees. Reducing browsing pressure contrarily led to tree establishment. They also found that large grazer populations, especially wildebeest and impala, exert significant pressure on grassland ecosystems directly through biomass removal and indirectly via grazer-grass-fire interactions. The effects of grazers tempering wildfires by removing potential fuel is often described alongside the control over the amount of organic matter biomass and nutrients entering the soil pool (Schmitz et al., 2014). Pachzelt et al. (2015), who also used LPJ-GUESS in combination with a grazer-specific population model, found that a high grazer density asserts a substantial impact on grass biomass, tree biomass and burned area. They found herbivory-induced responses in vegetation mass is similar to the responses found in our simulations (in our analysis here we did not specifically analyse impacts on fire). The importance of grazers is also highlighted by Kiffner and Lee (2019), who showed

that herbivore grass consumption can triple browse consumption, as reported in Lake Manyara Nation Park in Tanzania. However, the coupled model system so far does not differentiate herbivores into grazers and browsers.

## 4.2 Evaluation of the coupled model system

Human activities, including timber harvesting, livestock grazing, and in general disruption of natural trophic chains, are not captured by our simulations but are expected to be included directly and/or indirectly in large-scale empirical datasets we
used here for evaluation (Ripple et al., 2015; Wardle and Bardgett, 2004). These impacts can lead to either increases or decreases in total biomass compared to the hypothetical human-free world, which we simulate in this study. In Africa, for instance, the human pressure on dry forests and savannas is high due to the dependence of people on savanna ecosystem goods and services like timber for construction, fuelwood and charcoal, land surface for livestock grazing and wildlife tourism (Osborne et al., 2018). Such pressure likely explains in part the model's overestimation of the observed ecosystem's
carbon stocks throughout the African continent. Conversely, forest management across the forested areas of Europe (which are dominated by northern temperate and boreal forests) may not lead to reductions in total biomass. Lindeskog et al. (2021) for instance found little difference in simulations of forest above-ground carbon with and without wood harvest in LPJ-GUESS. Erb et al. (2018) also found reductions in aboveground biomass stock in used tropical and subtropical forests and savannas, to be substantially larger compared to reductions in used boreal forests - and also larger when compared to
managed temperate forest, even though the differences were less pronounced.

Additionally, the significant impact of human activity on trophic chains complicates the comparison of our results, particularly when comparing results with and without trophic chains in an otherwise undisturbed vegetation. Large-scale ecosystem disruptions have included widespread reductions in apex predators such as lions, cougars, and wolves (Morris and Letnic, 2017; Terborgh et al., 2001), cascading effects on large herbivore populations (Holdo et al., 2009), and the exclusion
of smaller predators like foxes through fencing for livestock protection (Croll et al., 2005). Despite local variations, the long-standing scientific consensus holds that these disruptions to the trophic chain contribute to the overall degradation of worldwide ecosystems and impact global vegetation biomass (Estes et al., 2011).

Confidence in overall model performance is bolstered by the good fit against the estimates by Pugh et al. (2023), which were derived for areas with relatively little direct human impact. Still, these estimates are strongly model-based and include
simulations from LPJ-GUESS with an adjusted natural disturbance function compared to our model version. This methodological overlap likely contributes to the observed agreement. Furthermore, the estimates by Pugh et al. (2023) are approximately 30% lower than those reported by Thurner et al. (2014), who estimated low uncertainty regarding human influence in their dataset. Despite this discrepancy, the results from both Thurner et al. (2014) and Pugh et al. (2023) align closely with our simulation outputs.

Generally, vegetation datasets for Europe match the simulation results better than dataset for Africa. As mentioned above, the degradation of savanna biomes due to the impacts of human activity might have had more drastic effects on total

vegetation biomass when compared to forest management in European forests. This difference could help explain the stronger agreement between our simulations and European datasets.

The power-law relationships derived from simulations of the various Madingley model versions suggest that the "online" version best represents animal population responses to underlying vegetation productivity. We draw this conclusion since the "online" version's power-law slope aligns best with the empirical relationship reported by Cebrian (2004), whilst maintaining a realistic representation of herbivore consumption rates. However, a notable overestimation of herbivore biomass, as first highlighted in Figure 5 of Harfoot et al. (2014), remains evident.

Our study results show that including herbivory via the coupled model system still upholds Madingley's and LPJ-GUESS's ability to produce realistic biome plant species distributions, carbon fluxes and animal population distributions. Nevertheless, it is important to acknowledge that the empirical measurements underpinning these comparisons likely reflect human influence, which is not accounted for in the Madingley model.

## 4.3 Sensitivity of results to important model assumptions

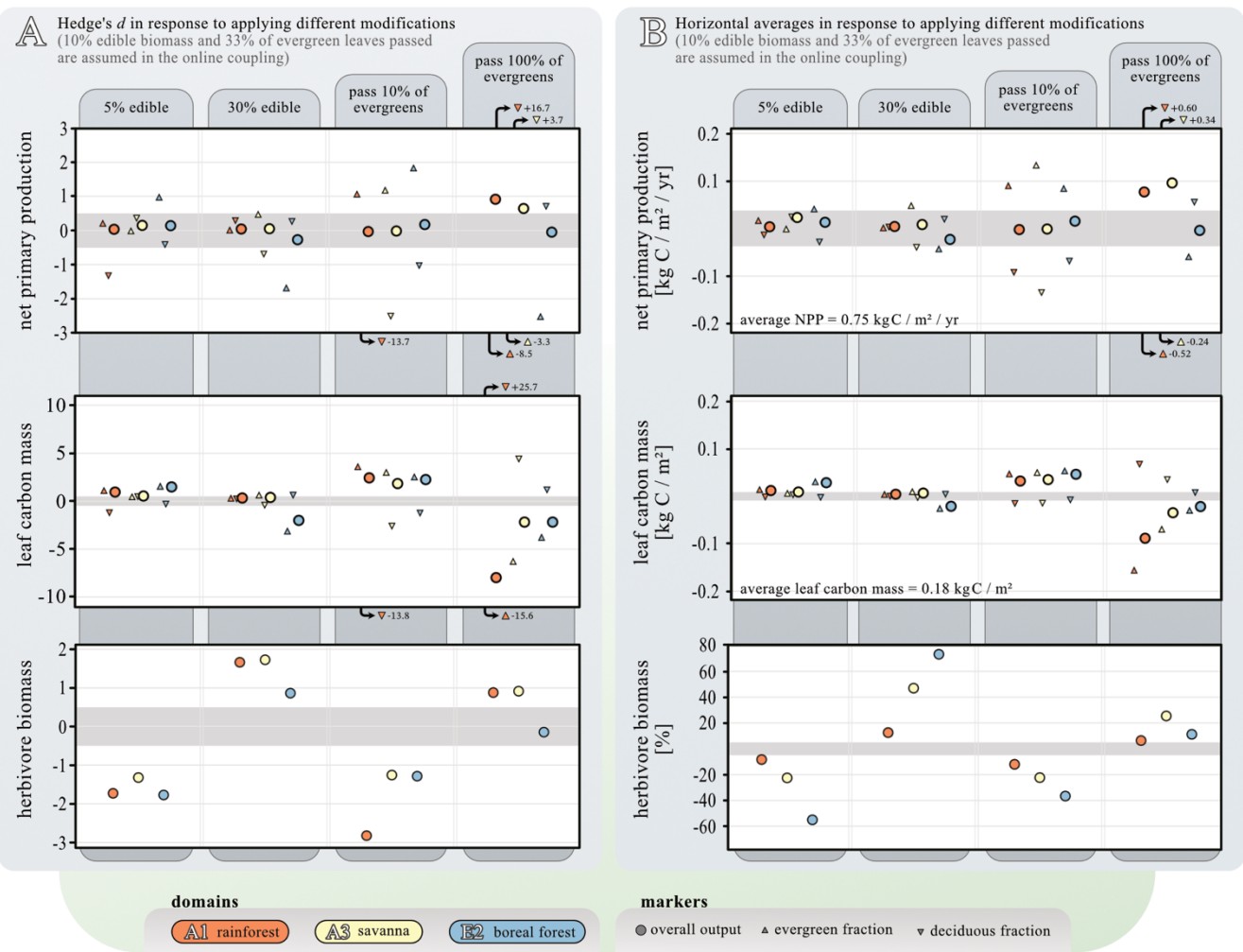

Figure 11: Sensitivity study of two model modifications: the amount of biomass edible for animals and evergreen vulnerability. The analysis was carried out for the simulation years 1900–1980. Both panels show data from three selected locations: orange – AF1 rainforest, yellow – AF3 savanna, and blue – EU2 boreal forest, each comprising 25 grid cells. Upward triangles indicate data of evergreen PFTs, while downward triangles indicate data of deciduous PFTs. Circles indicate data aggregated across all PFTs. Panel A shows the general effect size (Hedge's $d$). This data's 95% confidence intervals are similar within each metric: NPP ($\pm0.3$ SD), leaf carbon mass ($\pm0.4$ SD), and herbivore biomass ($\pm0.2$ SD). Small effect sizes are assumed for Hedge's $d<0.5$ (indicated by gray background shading). Panel B shows changes in domain-wide averages in response to the same two modifications. To set absolute values into perspective, the average NPP and leaf carbon mass across all three model domains and ensembles is given. Average responses smaller than 5% are indicated by gray background shading.

We explored the impacts of varying (i) the amount of edible biomass and (ii) reducing evergreen responses to biomass removal by analysis of the effect sizes using Hedge's $d$ (Figure **11**). Altering the proportion of edible biomass (from the 10% baseline assumption) has a relatively small impact on NPP and leaf biomass. Notably, only increasing the edible biomass to

30% in the boreal ecosystem leads to a significant decline in leaf biomass. This can be attributed to the fact that it is the only ecosystem where growth appears to be constrained primarily by limited forage availability, rather than by predation pressure. The responses in average NPP and leaf biomass are also small (<5%) across all ecosystems when varying the proportion of edible biomass, but with outliers visible for the boreal evergreen NPP in the 30% experiment (analogue to leaf biomass) as well as when reducing edible biomass to 5%. Herbivore biomass responds strongly to both reduced and enhanced available food.

Reducing the proportion of leaves passed from evergreens to Madingley from 33% to 10% (as stated by our baseline assumption) results in an increase in evergreen NPP at all three locations and a decline in deciduous NPP. The overall impact on NPP is small. As expected, evergreen leaf biomass increases, compensating for the decline in deciduous leaf biomass. This is evident in the overall increase in leaf biomass, which is most pronounced at location A1. Conversely, making 100% of evergreen leaves available to herbivores in Madingley results in notable increases in NPP – especially in the rainforest and savannah locations – as well as declines in leaf biomass across all locations. This response is primarily due to increased light transfer throughout the canopy and a shift in vegetation structure from predominantly evergreen (with ~90% evergreen leaf biomass) to largely deciduous/herbaceous (with ~30% evergreen leaf biomass). This corresponds with a significant reduction in evergreen NPP and leaf biomass (Hedge's d of -8.5 and -3.3, respectively), as well as an increase in deciduous leaf biomass (Hedge's d of +16.7 and +3.7). Herbivores respond to this change as if there were an increase in the availability of edible biomass, as they experience greater availability of leaf biomass. The sensitivity experiments show that varying the proportion of edible biomass has limited effects on overall NPP and leaf biomass across ecosystems, except under extreme scenarios—where significant shifts in vegetation structure and herbivore dynamics occur. These findings highlight the sensitivity of evergreen-dominated ecosystems to changes in foliage availability, with pronounced feedbacks on plant composition, light dynamics, and trophic interactions.

## 5 Summary

In this study, we examined the impacts of herbivore feedback on vegetation dynamics simulated in LPJ-GUESS by comparing the "default" version of the model with the "online" coupled version. The "online" simulations, which incorporate bidirectional feedback between vegetation and herbivores, resulted in a domain-wide reduction of ecosystem net primary productivity (NPP) by -5%, leaf area index (LAI) by -9%, and vegetation carbon mass by -10%. These effects were most pronounced in the boreal domain, where vegetation carbon mass decreased by -42%.

We also analysed animal population dynamics by comparing the "default" and "online" versions of Madingley. The "online" version showed reduced biomass density across all functional groups, driven by an overestimation of NPP (by "default" Madingley) throughout most of the domain. However, in regions where LPJ-GUESS simulated high NPP, the "online" herbivore populations are similar to those of the "default" version.

Evaluation of output from both the "default" and the "online" version of LPJ-GUESS against remote sensing datasets and flux measurements highlighted that the coupled LPJ-GUESS/Madingley model preserves LPJ-GUESS's ability to predict realistic biome distributions and carbon pools. We also compared power-law relationships of herbivore population dynamics for all three different versions of the Madingley model and concluded that the "online" version surpasses both other versions in terms of representing similar power-laws derived from empirical data.

With the coupled model system in place, we are now in a position to explore different use cases of such a model system. One of those cases is presented in the following Chapter in the form of investigating the removal of large animals and ecosystem capabilities to recover from such removal.

## 6 Limitations and Future Development

The bi-directional coupling of LPJ-GUESS and Madingley is a further step towards exploring how plants and animals interact in natural and human-modified ecosystems and regulate biogeochemical cycling. A number of important processes are still lacking that prevent us from doing this holistically.

For example, this version of the coupled models is lacking explicit C:N cycling through animals. Implementing C:N stoichiometry into the process descriptions would enable herbivores to (i) choose their diet, which would affect quantity and type of biomass consumed and (ii) return animal-based litter with the relatively enriched N content found in nature, which is expected to affect nitrogen cycling in soils. A related development is the need to differentiate between grazers and browsers in Madingley and to incorporate proper sub-annual allocation of carbon and nitrogen to tissue growth in LPJ-GUESS.

The current version of the model system also preserves damage to evergreen leaves longer than damage to deciduous leaves. While evergreen plants normally invest more resources into long-lived leaves in the form of defence mechanisms (Coley and Aide 1991), the absence of such mechanisms in the modelled world is creating a competitive disadvantage for evergreen PFTs through an overestimation of herbivore consumption of their leaves.

The presented simulations mimic a pristine world without human impact. LPJ-GUESS is part of the Land-SyMM framework (www.landsymm.earth), which enables simulations of human land management and related land cover and land use change. In the future, we aim to include the implemented coupling between the Madingley model and LPJ-GUESS into Land-SyMM framework and so begin to explore mechanistically interactions between humans and ecosystems of interacting plants and animals.

## 7 Code and data availability

The LPJ-GUESS model code is available under the open access repository (https://doi.org/10.5281/zenodo.11401444). The original Madingley model code is managed by the UN Environment Programme World Conservation Monitoring Centre (UNEP-WCMC) at the University of Cambridge, England. The intellectual property right for the translated version we used

in this study is hold by the Radboud University in Nimwegen, Netherlands. Therefore, a DOI for the Madingley model code cannot be provided publicly. The source can be made available under a collaboration agreement under the acceptance of certain conditions. The input data used for the simulations are publicly available and cited in the manuscript.

Output data from the simulations are available under the open access repository (https://zenodo.org/records/12788281).

## 7 Author Contribution

Jens Krause: Conceptualization, Methodology, - Writing, Original draft, Visualization.

Mike Harfoot: Conceptualization, Methodology, Software, Writing – Review & Editing.

Peter Anthoni: Software, Validation - Review & Editing.

Moritz Kupisch: Conceptualization, Methodology

Almut Arneth: Validation, Writing – Review & Editing, Supervision, Funding acquisition.

## 8 Competing Interests

The contact author has declared that none of the authors has any competing interests.

## 9 Acknowledgement

Almut Arneth, Peter Anthoni and Jens Krause acknowledge funding via the Helmholtz Foundation Programme, Changing Earth, Germany. We also thank Matthew Forrest from the Senkenberg Institute Frankfurt for providing the above-ground biomass dataset for non-cropland and non-pasture vegetation.

## 10 Statement on AI-Assistance

We used OpenAI's ChatGPT v4 and Grammarly v1.100.2.0 to assist with text rephrasing and linguistic refinement. The AI tool was employed exclusively to improve the clarity and coherence of an original manuscript, while the intellectual content, analysis, and conclusions remain entirely my own. Subsequently, the text was critically reviewed and further edited. All use of AI adhered to ethical guidelines and institutional policies regarding the use of technology in academic work.

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
