# Peer review of "Modelling Herbivory Impacts on Vegetation Structure and Productivity"

_EGUsphere, 2024_

## Author Response (AR1)

Dear Editor,
Dear Reviewers,

Thank you for the opportunity to revise and resubmit our manuscript titled "**Modelling Herbivory Impacts on Vegetation Structure and Productivity**" (egusphere-2024-1646), submitted to Geoscientific Model Development. We appreciate the constructive feedback provided, which has helped us improve the clarity and quality of our work.

We have carefully addressed all reviewer comments and, in response to them, made substantial revisions throughout the manuscript. We revised the abstract to better reflect the key findings of our study. The model descriptions were expanded to clarify technical aspects raised by the reviewers. We also thoroughly restructured the Results and Discussion sections to enhance readability and improve the logical flow of arguments. Some figures were completely redesigned to present more informative and thematically cohesive content. We now also report coupling-related changes across all trophic levels in Madingley, and for key locations, we provide more detailed comparisons with results from the previously published "offline" coupling approach (Krause et al., 2022).

We have conducted a sensitivity analysis at three representative sites. In these simulations, we varied (1) the amount of biomass available for herbivores and omnivores and (2) the proportion of evergreen leaves passed to Madingley. We believe these two factors represent the most influential assumptions governing the behaviour of the coupled model system.

We would also like to disclose that we used AI-based language assistance (ChatGPT 4.0 by OpenAI and Grammarly V1.100.2.0) to help revise the manuscript for clarity and language quality. This use is transparently acknowledged in the manuscript.

A detailed, point-by-point response to each reviewer comment is provided below. We submit two version of the revised manuscript, a clean one, and one with changes made indicated. The line numbers indicated in the table below refer to this marked-up version of the manuscript.

We hope that the revised manuscript meets the expectations of the reviewers. Thank you again for your time and consideration.

Sincerely,
Jens Krause, on behalf of all authors

| Comment of reviewer 1 | response / manuscript changes | lines |
|---|---|---|
| *"As the first try, I understand the complexity (and difficulty) of coupling two well-established models, and why the authors chose to use the data files to exchange the information between these two models and the interactions between vegetation biomass and herbivores. However, I still want to see more details of the model mechanisms and processes that drive the vegetation and trophic dynamics through animal foraging."* | We acknowledge that providing more details about the processes in both models further enhances the understanding of the coupling mechanism and its effects. In the revised manuscript, we have extended the model descriptions for both Madingley (2.1) and LPJ-GUESS (2.2) that are particularly relevant to our study. For Madingley, we included critical detail about the basic assumptions for the feeding methodology, animal interactions, reproduction, and dispersal. For LPJ-GUESS, we added information about annual plant growth and allocation of carbon. | 100 – 156 |
| *"As I can get from the method section, the browsing of herbivores from the Madingley model acts as a disturbance to vegetation leaf biomass. So, I think it would be necessary for the authors to give more details how the demographic and growth processes respond to the reduction of leaves. For example, do they have a "compensation" mechanism to grow new leaves? Are all the trees affected by animal foraging samely (i.e., the same leaf reduction ratio) or these effects are size-dependent? (because the taller trees have low probability of being eaten by animals compared to the seedlings). Also, how is the size structure and regeneration of vegetation are affected?"* | We elaborated on the related growth process in the LPJ-GUESS model description, as well as in the description of the coupled feedback link.
Plants do not have an inherent "compensation" mechanism, since LPJ-GUESS does not regrow leaves throughout the year; carbon allocation is modelled as an annual process.
All PFTs are affected equally by the reduction, but their current phenology state is taken into account (described now in the supplementary material). This ensures that only PFTs that contributed towards the transferred leaf stocks are affected by the removal. Tree height or age is neglected when applying herbivory, as Madingley does not include a vertical structure at the current stage of development and PFT leaf masses are aggregated towards grid cell averages. The reviewer is correct in that these effects do occur in reality and they will be included in future model development. See section 6 "Limitations and Future Development". | 149 – 152
SM. 35ff |
| *"In the model, there should have at least two types of key parameters, consumption coefficients and regrowth rates, that are used to describe browsing effects and vegetation responses. More details are needed."* | Consumption coefficients (here called assimilation efficiency) are provided for herbivores and omnivores in Table S2. We also added the exact values to the manuscript for clarification. Regrowth rates are not prescribed in LPJ-GUESS. A PFT age-cohort sheds a fraction of their leaves (100% if deciduous, 33% is evergreen) every year. It then uses a part of the accumulated annual NPP to grow new leaves. We have added a clearer description of this process in the LPJ-GUESS model description. | 119
Tab. S2 |
| *"The authors analyzed the vegetation productivity and biomass changes in response to animal foraging. For the trophic model (Madingley), it should have the biomass (or density) of herbivores, omnivores, and carnivores. Is it possible to present their biomass/density as well? I think it would be informative for understanding how the coupled models work and to justify if the results are realistic, because I think the effects of herbivores on forest ecosystems are overestimated in this model. It like surface fire in savanna, once the tree seedlings escape the "browsing trap", they will not be threatened by animals."* | We have added a new figure that describes coupling-related responses if biomass density of the different trophic levels in great detail. This new figure 6 includes both maps of the overall model domain and detailed comparisons at different locations.
We explore how different degrees of leaf biomass removal affect vegetation in a newly added experiment and section in the discussion. The 'browse-trap' the reviewer mentions is indeed an important process in savannas – as mentioned above at this point in time, Madingley does not include a height-resolved feeding, thus seedlings are not specifically targeted by herbivory. Furthermore, LPJ-GUESS is handling seedlings as a reproduction/ establishment pool. We add this to the revised discussion. It is -as indicated in our earlier response- an important next development step. | Fig. 6
535 – 558
651 – 654
666f |
| *"So, I think it needs a comprehensive uncertainty analysis in the mode-data evaluation with a ensemble runs at different key parameters."* | We have performed a sensitivity analysis for the major assumptions (the amount of biomass edible for herbivores and the fraction of evergreen leaved passed to Madingley) to test our coupling assumptions. In this sensitivity study, to ensure stochastic stability, we ran an ensemble of 3 simulations, which all deviated less than 1% from another. | 462 – 471
Fig. 11
732 – 755 |

| Comment of reviewer 1 | response / manuscript changes | lines |
|---|---|---|
| *"Lines 67~82, about the Madingley model, it would be easier for the readers if the consumption rate and energy flow ratio from vegetation to carnivores are described."* | The Madingley model only takes into account wet matter assimilation. Consumption rates are not part of the standard output of the Madingley model. The model does only assume certain assimilation efficiencies for each AFT. | Tab. S2 |
| *"Lines 98 ~ 122, LPJ-GUESS model, I'd like to see how the demographic processes and cohort dynamics are simulated in LPJ-GUESS, because they can be affected by herbivore foraging."* | As mentioned above, we have elaborated on the processes affecting PFT growth in the LPJ-GUESS model description. | 149 – 152 |
| *"Lines 140~141, why the evergreen leaves have longer damage effects than deciduous leaves? Why is it related to the leaf lifespan? New leave can grow out anyway."* | Leaves are regrown on the first day each year in LPJ-GUESS. Deciduous trees typically have a leaf turnover rate of 1, thus shedding (and regrowing) all of their leaves. Evergreen PFTs typically have a leaf turnover rate of 0.33, thus shedding one third of their leaves every year. Therefore, the herbivory damage to the remaining two-thirds of the leaves is conserved into the following year.
We are aware that this is a limitation of the model and a new, sub-annual carbon allocation is currently in development. We added a paragraph to the limitations to clarify this limitation. | 782, 783 – 786 |
| *"Lines 145~150: I am confused here. In the previous section, the authors talked about the damages to leaves. However, here, they talked about the biomass. Does it only include "leaves"? Or, the branches and stems are included?"* | We are sorry for the confusion. In Madingley, the vegetation stocks are always leaf mass only. The reduction is in fact only applied to leaves. To clarify this, we changed the description to "leaf biomass reductions", while also shifting this section to the supplementary material. | SM. 33f |
| *"Section 2.5 Study setup. I think it would be helpful if the authors conduct a set of ensemble runs with different parameter values at site level and then report the changes in the biomass at different trophic levels. The large spatial scale runs cannot give such details."* | We have included this in the sensitivity study, that is added at the end of the discussion section. | 462 – 471
Fig. 11
732 – 755 |
| *"Line 215, Figure 3. The legend of the vegetation types can be the full name. there are enough spaces for that."* | We have redesigned every non-evaluation figure in the manuscript to focus on the most relevant information. The different PFT dominance patterns are now part of Figure 5 and we have added full names for the different PFTs. | |
| *"Lines 229~231, Also, I think it is extremely high for the browsing effects on boreal forests. That is why we need to conduct site-level runs with different parameter values. Or, the setting of eatable evergreen leaves needs to be examined carefully."* | Runs with different assumptions for edible biomass fractions at the boreal EU2 location are included in the sensitivity study. We indicate the larger effect size to modifying the baseline 10% proportion of edible biomass in the discussion section. | Fig. 11
732 – 740 |
| *"Line 238 "no shifts in vegetation compositions". Savanna can be affected by browsing. Are the PFTs and different sized trees are uniformly affected by animals?"* | Yes, they are affected uniformly. We are aware that this is a major limitation, as well as the lack of a differentiation between grazers and browsers. | 679
781 |
| *"In Discuss section, I think the interactions between demographic processes and the herbivore browsing should be explained."* | This is included in the revised discussion section. Especially the start of section 4.1 covers interactions and expected impacts of animals on photosynthesis. | 637ff |

| Comment of reviewer 2 | response / manuscript changes | lines |
|---|---|---|
| *"First, in contrast to the earlier work (Krause et al.), it is important in the coupled version that the processes simulated in Madingley are appropriately fed forward and fed back. Now, Madingley has things such as animal movement, but LPJ-GUESS is running on a 0.5° (lat, lon) grid, and hence spatially explicit effects from Madingley are neither driven correctly by LPJ-GUESS, nor can the feedbacks be treated appropriately in LPJ-GUESS (this is not a criticism, but just a fact). How important are these processes in Madingley, and what are the implications (possible bias) when they are ignored or misrepresented in the present model application? In other words, how can the authors assure that the two-way coupling is not leading to 'pathological' feedback that leads to bias in the simulation results? For example, the huge impact of moose in boreal forests appears grossly exaggerated."* | The reviewer raises important questions, which we are happy to clarify in the revised version. We use for both models a 0.5° grid resolution, as previously described in Krause et al. (2022), but not in our manuscript. We now added relevant information also into the revised methods section, since assuring spatial explicitness in the coupling is indeed important. In particular, the grid cell list that LPJ-GUESS needs to run, is generated by Madingley by listing every terrestrial grid cell coordinate and sending them to LPJ-GUESS afterwards. This assures that there are:
 A) No grid cells simulated by LPJ-GUESS that do not witness animal mediation and influence overall domain-wide averages.
 B) No grid cells in Madingley are uncoupled to LPJ-GUESS and are accidentally still powered by the build-in vegetation model. This is essential, as such a grid cell would act like a safe-haven for herbivores, as the build in vegetation model tends to model higher vegetation levels and knows no feedback link between the vegetation growth and herbivory
 We also adapted the text describing the overall coupling loop to assure the reader that dispersal (as singular process that can affect multiple grid cells) is handled after feeding is processed and the corresponding leaf damage is passed to LPJ-GUESS. Thus, dispersal does not break carbon mass conservation in LPJ-GUESS.
 We thank the reviewer for pointing this out and giving us the opportunity to clarify this technical demanding issue. | 103f
 219 – 224 |
| *"Second, it is not principally a problem that LPJ-GUESS is simulating potential natural vegetation, whereas most data sources used to benchmark the uncoupled and coupled models are from managed ecosystems. If a clear expectation was brought forward how the models should deviate from the data (also for which region – savannas and boreal forests may be subject to different expectations, for example), this would be quite helpful for the assessment. However, the manuscript does not actually contain a clear set of research questions at the end of the Introduction, but we are learning about the goals of the paper much later on (e.g. lines 287, 303, 359-360)."* | We have modified the end of the introduction to clarify the goal of this study.
 In our view, the impact of implementing herbivory (reducing vegetation biomass) is not necessarily expected to compensate for the potential overestimation resulting from the exclusion of human influence in both versions of LPJ-GUESS. This is partially because herbivory affects 'only' leaf biomass, whereas human influences (timber harvest etc..) affect vegetation structure much more strongly. Therefore, we expect both model versions to overestimate empirical data. We address this expectation in line 602.
 In the revised discussion section, expectations about the coupling-related trends in vegetation biomass are included in section 4.1 (l. 644ff). | 79 – 81
 602
 644 – 646 |
| *"Third, although the paper is about the coupling of LPJ-GUESS with Madingley, not a single result from that latter model is shown. All we are presented with are results of LPJ-GUESS. However, in a coupled system it would be important to learn about both counterparts. The manuscript would gain a lot if also the animal side of things was presented for the European and African domains, and put into context with the available literature on tropic chains in the different biomes/ecosystems. This would also be an important element to learn about and better understand the results of LPJ-GUESS (cf. the moose example further above – is it true that Madingley is simulating large amounts of _this_ Animal Functional Type indeed?)."* | Fair point, thanks for highlighting this. We have added a new figure that describes coupling-related responses if biomass density of the different trophic levels in great detail. Said figure includes both maps of the overall model domain and detailed comparisons at different locations.
 In the revised discussion section, we highlight the special role of omnivores when it comes to explaining the large impacts of herbivory in boreal ecosystems. | Fig. 6
 535 – 558
 651 – 654 |

| Comment of reviewer 2 | response / manuscript changes | lines |
|---|---|---|
| *"Fourth (and least importantly), the writing of the paper needs to be improved considerably. I have already noted the absence of research questions (or hypotheses, or goals). Also, the organization of the text, particularly in the Results and Discussion sections, is not satisfactory. Paragraphs contain multiple topics, and the reasoning is going back and forth. This makes for very rough reading at least in portions of the manuscript. Lastly, the manuscript needs careful checking for grammar mistakes and general "elegance" of writing. It is a rough piece at the moment. More on this below."* | We have thoroughly revised and reorganized the Results and Discussion sections to improve clarity and readability. Figures 3 to 6 were updated to include more relevant information that supports the new structure of the Results section. We begin by addressing the general impacts and emerging trends related to the model coupling, with a focus on the underlying processes driving differences between model versions and whether these trends align with findings from the literature. Subsequently, we reflect on the evaluation of the coupled system, considering the absence of human influence in the simulations and its implications for interpreting the results. Finally, we incorporated the sensitivity analysis at the end of the Discussion to address potential reader concerns regarding the influence of our assumptions on model output. | sec. 3 & 4 |
| *"Abstract: contains too little on the results of the coupled model (essentially just two lines, 15-17). The last sentence (which should be a conclusion) brings a surprising statement: the goal of the entire exercise was that LPJ-GUESS should still perform well? This cannot be. Hence the Abstract needs considerable work (shortening of first part, more results, and elements of discussion/conclusion to be brought out)."* | The revised abstract now includes a larger proportion of the study results – including the main trends simulated for NPP, GPP, LAI and vegetation carbon. We also mention the observed increase in light transfer throughout the canopy in response to the coupling. | 15 – 20 |
| *"25-28: These three hypotheses are not mutually exclusive, but they are presented here as a "debate". I think this is misleading."* | We have revised the sentence. | 26 – 29 |
| *"33-45: Nice review of the (limited) literature on the subject. The next sentence (lines 46ff.) then raise the expectation that we will learn something how these interactions reverberate to the entire trophic chain. However, the entire paper does not contain a single hint to this... either remove such expectations or expand the paper considerably (cf. my general comments)."* | We have retained the sentence in the manuscript, as it highlights a significant limitation in many current ecosystem modelling approaches. This point also serves as a key justification for selecting the Madingley model as our coupling candidate, given its capacity to represent such interactions. Such interactions are included in the presentation of animal responses to the coupling. However, a comprehensive exploration of food web reverberations will be addressed in a future publication, as it falls beyond the scope of the present study. | 70 |
| *"52-53: This reference needs to be fixed. Generally, the parentheses are often not correct with literature citations (e.g. lines 81-82, but also in many other places). Check the reference manager."* | | 76 |
| *"68-82: I have elaborated on the scaling problem in the coupling between LPJ-GUESS and Madingely above. As a matter of fact, the issue is hidden rather well in the manuscript. We learn only much later (l. 191) that LPJ-GUESS is running on a 0.5° grid. I think the authors should be more explicit and exhaustive in the technical description here, and address the scaling problem."* | See our response to this issue above. We are confident that the coupling is functioning as intended. However, we acknowledge that it was not clearly communicated that both Madingley and LPJ-GUESS operate on the same spatial grid. We have revised the manuscript accordingly to clarify this important detail. | 103f 219 – 224 |

| Comment of reviewer 2 | response / manuscript changes | lines |
|---|---|---|
| *"93-98: This is technical jargon that is very hard to understand for "outsiders"; for example, why is the ability to switch the "temporal and spatial grid cell handling loop" (what on Earth is that??) necessary? I was totally lost with sentences like this one. Then comes the reference to animal migration (at 0.5 °resolution? I know that there was an attempt to parameterize this, albeit for seed dispersal, by Snell et al.; however here we are dealing with another issue). Next, why is monthly "herbivory-driven defoliation" needed in LPJ-GUESS? This remains unclear. This description requires a much more careful treatment and more detail."* | We have added a detailed explanation of how LPJ-GUESS typically handles spatial and temporal loops, and clarified why we modified this sequence to enable effective coupling with Madingley. We hope the revised description improves clarity and accessibility. More broadly, we have also worked to reduce technical jargon throughout the manuscript to enhance overall readability.
 We originally mentioned migration in this context because synchronizing the model domain to operate on the same time step is crucial for processes that span multiple grid cells—something the original LPJ-GUESS framework could not accommodate.
 Seed dispersal is currently not represented in the coupled system. LPJ-GUESS does not simulate seed-based reproduction, and Madingley does not resolve the small-scale movement patterns required to model seed dispersal effectively. | 164 – 218 |
| *"99-107: The description of this (very complicated) spin-up system is not explained well here, and it is not congruent with Fig. 1, neither in terms of the terminology nor the time scales (e.g., a 10,000-yr period is mentioned in the text, but cannot be found in Fig. 1). This must be improved."* | We have isolated the model spin-up description in subsection 2.4 to clearly isolate and describe the model spin-up procedure, making it easier for readers to focus on this important aspect of the methodology. Additionally, we removed the reference to the 10,000-year soil spin-up, as this study does not focus on soil processes, and no changes were made to the default LPJ-GUESS soil spin-up configuration.
 To improve clarity, we also separated the runtime illustration from the technical coupling schematic. The revised Figure 1 now presents only the technical structure of the model coupling, while the new Figure 2 illustrates the modified runtime workflow of the coupled simulation. | Fig. 1
 Fig. 2
 228 – 244 |
| *"111: Over time, the reader starts to realize that the link to higher tropic levels is via leaf biomass; there are no other herbivory effects considered (note that functionally, animals like the Mountain Pine Beetle are herbivores as well). I think it would be valuable to state this in the Introduction already, to avoid false expectations."* | We added this to the introduction, as well as the abstract to ensure this is communicated earlier in the manuscript. | 14
 81 |
| *"114: The reference to Fig. 1 is coming too late; and the text on lines 108-114 is quite confusing and hard to understand."* | We mention Fig. 1 earlier in the revised section 2.3. | 218 |
| *"118-120: It seems a very poor assumption that all carbon consumed by herbivores is returned to the litter pool immediately. A more reasonable assumption would be that 90% of the C is respired by the herbivores, and only 10% is returned to the litter pool. What is the substantiation for this entirely unrealistic assumption (which may have considerable implications for the simulation results)?"* | We included a test of this assumption in the sensitivity analysis presented in the supplementary material. Specifically, we conducted simulations in which only 10% of the consumed carbon and nitrogen was transferred to the corresponding litter pools. The results indicated that this modification had no significant impact on the overall model outcomes.
 Based on the sensitivity study, we chose not to alter our 100% transfer assumption in the main simulations, as our focus was on the isolated effects of aboveground biomass consumption. Looking ahead, we plan to couple LPJ-GUESS with a nitrogen-enabled version that tracks nitrogen dynamics through Madingley's ecological processes. This will allow us to more accurately represent nitrogen and carbon contributions to the litter pools, including inputs from animal faeces and carcasses. | SM:
 56 – 73 |
| *"125-129: I am not convinced that the description of how exactly the data exchange between the models is done is necessary in the main text. This would be something for the supplement. I think the main text should focus on the conceptual things, rather than file details etc."* | In agreement with reviewer 1, who also suggested this change, we moved the detailed description of the coupling loop to the supplementary material. | |

| Comment of reviewer 2 | response / manuscript changes | lines |
|---|---|---|
| *"134: At this point, we have no idea what a "grid cell" is (cf. further above). Be more explicit and upfront with these (potentially important!) things."* | Grid cells are first described in the Madingley model description, as it is essential to the implemented migration process. | 104 |
| *"134-153: Most endothermic animals (such as moose in boreal forests) do not have access to most leaf biomass, because the point of trees is to grow tall (competition for light), which removes a lot of the ecosystem leaf area from the reach of ungulates. It appears that this is ignored when assuming that very sizeable fractions of leaf area are accessible to herbivores, such as one third of all evergreen leaf biomass. What are the effects of this (totally unrealistic) assumption? Could this be the cause that the model produces excessively high reductions of leaf area particularly for boreal forests? However, such effects should also be visible elsewhere in the simulated domain. This needs elaboration, explanation and clarification."* | The reviewer is absolutely right—this represents one of the most significant current limitations of the Madingley model. The assumption in question, originally introduced in Harfoot et al. (2014), leads to a structural bias that can disproportionately favour larger animals, as it does not account for vertical vegetation structure or realistic accessibility of biomass. To date, there have been no systematic assessments quantifying the specific impact of this assumption on simulated animal populations. Nevertheless, within the Madingley modelling community, there are ongoing efforts to improve this aspect of the model. These include incorporating vertical vegetation structure and developing more sophisticated representations of edible biomass fractions, which we believe will enhance the ecological realism of future model versions. We highlighted this assumption and its inherent limitation in the Madingley model description. | 112 – 114 |
| *"153: I am not sure that the coupling between the two models is done in a correct manner, from a systems theoretical point of view. It appears that both models are discrete-time models, and hence the state of L(t) [LPJ-GUESS at time t] should influence the state of M(t+1) [Madingley at the next time step], and NOT M(t), and vice versa. I am fully aware that reality is different, because reality does not know about a time step. Yet, model formalisms are clear and unequivocal (cf. Zeigler 1976, _Theory of modelling and simulation_. Wiley)."* | Thank you for your comment. As noted above, we are confident that the coupling is functioning as intended. However, the connection between model timesteps involves some complexity. In the current setup, leaf regrowth in LPJ-GUESS is calculated annually, at the beginning of each year. Meanwhile, leaf biomass is reduced monthly based on consumption by herbivores and omnivores in Madingley. This reduction influences potential future growth (which is simulated as daily increment), but does not feed back into the LPJ-GUESS allocation process until the next annual step. We acknowledge that this setup involves integrating processes across three distinct temporal resolutions (daily, monthly and annually). At present, full temporal synchronisation is limited by the annual allocation cycle in LPJ-GUESS and the monthly timestep in Madingley. A more harmonised coupling will be possible once LPJ-GUESS supports sub-annual allocation and Madingley is run at a finer (daily) timestep. Until then, this compromise remains necessary to enable coupling between the two models. | |
| *"192: It appears that this is a dataset that was _aggregated_ to 0.5°, not "downscaled"."* | Correct. We adapted the part of the manuscript. | 433 |
| *"210: As mentioned in the general comments, we learn nothing about the behaviour of Madingley in this manuscript. I do not think that this is acceptable, and it is certainly not helpful if we want to understand the (sometimes truly puzzling) behaviour of LPJ-GUESS in coupled mode."* | See our response to the general comment. | |
| *"221: The text around here appears to have been copy-pasted repeatedly... there was no mentioning of boreal ecosystems earlier, but of Eastern Europe; hence this cannot be "again", can it? The text is hard to read and understand in this paragraph."* | The whole results section was revised and reorganised to increase readability and ensure a better text flow. | |

| Comment of reviewer 2 | response / manuscript changes | lines |
|---|---|---|
| *"233: The switch to autotrophic respiration in the middle of a paragraph is quite confusing. Something is wrong with the logic here. The preceding material should probably be joined with the paragraph before?"* | We removed the mention of plant respiration in the revised results section to maintain focus on GPP and NPP. Since NPP already accounts for autotrophic respiration (as it is derived from GPP minus plant respiration), we believe that explicitly referring to respiration is redundant in this context and could distract from the main findings. | |
| *"239: At least with the first occurrence of a PDF shorthand in the text, pls add "PFT" (on this line, after the parenthesis) to make it easier for non-specialists to understand."* | We added the full names of PFTs to the manuscript before using their abbreviations in parentheses. | 516 517 519 |
| *"244-246: Three examples of grammar errors ("in regards of" –> "with regard to"; "area, which" –> "area that"; "evapotranspiration rates … responds" –> put verb in plural). Check manuscript throughout."* | We have carefully revised the entire manuscript to correct grammatical errors and improve overall language clarity. | |
| *"250: Figure 2 is occupying a lot of space, but in the text very little is made out of it. Perhaps instead of showing all these details without really picking them up in the text, omit part of it and show Madingley results (cf. general comments and further above)?"* | We chose to retain the technical illustration in the main manuscript because we believe it provides valuable insight into the structure of the coupling loop and supports readers in understanding and referencing our approach. However, if the editor prefers, we are happy to move Figure 1 to the supplementary material. | |
| *"256: Area 5 is also showing rather large differences, not just Area 1 & 2."* | In this context, our original statement referred specifically to boreal ecosystems; however, the reviewer is correct in pointing out the potential for confusion. To clarify, we have now explicitly added the reference to "area 5" in the same sentence. | 573 |
| *"269-272: This is an intriguing finding that hopefully will be discussed later… however this is not done."* | LPJ-GUESS is known for a tendency to overestimate LAI, and this tendency is not specific to our coupled version. As such, we consider it to be an issue unrelated to the coupling presented in this study. For that reason, we did not explore it further in the discussion. However, we felt it was important to briefly acknowledge this pattern, as it is a noticeable outcome in the evaluation results. | |
| *"287: It is truly surprising to learn about a goal of the paper here, towards the end of the Results section (cf. general comments). Make these things obvious as research questions (or so) at the end of the Introduction."* | See our response to the general comment. | |
| *"315-341: This portion of the Discussion is not knitted well together; the text is often rambling and the logic hard to follow. Please re-consider (hints are below)."* | We have restructured the discussion to enhance readability and ensure a clearer, more logical progression of the argument. | |
| *"315-317: Here the point is made that perhaps natural vegetation in temperate Europe was much more open before human influence started. Fine, but why then are the boreal systems showing the largest decrease in the coupled simulations? Worse, on line 321 the point is made that human influence did _not_ "drastically" change total biomass, which is in direct contradiction to the earlier statement. So what should we take away from this? On line 318, the argument switches away from this and focuses on the quality of the validation data, to return back to the issue of biomass reduction. This is hard to follow and confusing at best."* | We aimed to highlight how human influence can vary across different ecosystem types and historical contexts. We acknowledge that the previous representation lacked clarity and may have caused confusion. In the revised manuscript, we have improved this section by restructuring the argument to more clearly differentiate between the European and African ecosystem responses. | Sec. 4.2 |
| *"325: This could be a substantial artifact deriving from the way the coupling of the two models was done (cf. comment on lines 135ff.). So again, we need to know something about the biomass of herbivores and the different AFTs composing them, so as to be able to assess whether this simulated response makes sense."* | We have moved this section to Discussion subsection 4.1. This specific observation is now discussed in the context of the unique role omnivores play in the coupled model system, particularly their ability to adapt to forage shortages. In the "online" coupled version of Madingley, we observe an increase in omnivore biomass density, which reflects this adaptive capacity. | 652 – 654 |

| Comment of reviewer 2 | response / manuscript changes | lines |
|---|---|---|
| *"326: "combination of photosynthesis rate: with what? This is not stated."* | In the original manuscript, we intended to highlight the combined effects of leaf damage and the shorter growing season. We apologise for the mistake as we revised the explanation accordingly to ensure better understanding. | 649 – 650 |
| *"361: I would not combine limitations with the outlook (Conclusion?) of the manuscript. Rather have a section that deals with methodological limitations, and then add a section Conclusions that brings home the major messages of the paper (based on the research questions, which currently are lacking)."* | We have added a summary section (Section 5) to the manuscript. If preferred by the editor, we are happy to rename it "Conclusion."

We believe it is important to include a forward-looking section on future model development, as the current state of the model remains under active development and still includes some of the limitations previously discussed. | 756 – 773 |

---

## Author Response (AR2)

Dear Editor,
Dear Reviewer,

We thank you again for the opportunity to resubmit our manuscript with minor changes. We greatly appreciate the constructive feedback and the valuable contextualization of our work within the broader scientific framework.

During the revision process, we carefully considered all suggestions and made the corresponding modifications throughout the manuscript, while also incorporating the recommended minor changes.

The reviewer emphasised the importance of early foundational studies in the field from 1973 to 1992. We fully agree on the relevance of this literature, and we did include a citation to Dyer et al. (1993) who reviewed this body of literature (adapted in lines 418-421).

Furthermore, the reviewer noted that a description of the mathematical foundation of trophic interaction is not described in the manuscript. Such an in-depth description would exceed the scope of this study. However, in the original version of the manuscript, it was unclear where this information can be found. We now modified the end of the general Madingley model description (line 103ff) and refer readers directly to Harfoot et al. (2014) and the corresponding supplementary material.

Finally, the reviewer pointed out that not all acronyms were explained in the manuscript, which is why we added the full text formulation of "NPP", "GPP" and "AFTs" to the manuscript (lines 73f and 248f). We think that the acronyms are now fully documented - besides database names like "FLUXNET".

Thank you for your consideration of this manuscript,
Sincerely,

Jens Krause on behalf of all authors